# Evaluation of OMPS/LP Stratospheric Aerosol Extinction Product Using SAGE III/ISS Observations

Zhong Chen[1,2], Pawan K. Bhartia[2], Omar Torres[2], Glen Jaross[2], Robert Loughman[3], Matthew DeLand[1], Peter Colarco[2], Robert Damadeo[4] and Ghassan Taha[5]

[1]Science Systems and Applications, Inc., Lanham, MA, USA
[2]NASA Goddard Space Flight Center, Greenbelt, MA, USA
[3]Department of Atmospheric and Planetary Sciences, Hampton University, Hampton, VA, USA
[4]NASA Langley Research Center, Hampton, VA, USA
[5]GESTAR, Columbia, Maryland, USA

Correspondence to: Zhong Chen (zhong.chen@ssaihq.com)

**Abstract**

The Ozone Mapping and Profiler Suite Limb Profiler (OMPS/LP) has been taking limb-scattered measurements from April, 2012–present. It is designed to produce ozone and aerosol vertical profiles at 1.6 km vertical resolution over the entire sunlit globe. The Version 1.5 (V1.5) aerosol extinction retrieval algorithm provides aerosol extinction profiles using observed radiances at 675 nm. The algorithm assumes Mie theory and a gamma function aerosol size distribution for the stratospheric aerosol that is derived from Community Aerosol and Radiation Model for Atmospheres (CARMA) calculated results. In this paper, we compare V1.5 LP aerosol profiles with SAGE III/ISS solar occultation observations for the period June 2017 – May 2019, when both measurements were available to evaluate our ability to characterize background aerosol conditions. Overall, LP extinction profiles agree with SAGE III/ISS data to within ±25% for altitudes between 19 and 27 km, even during periods perturbed by volcanic eruptions or intense forest fires. In this altitude range, the slope parameter of linear fitting of LP extinction values with respect to SAGE III/ISS measurements is close to 1.0, with Pearson's correlation coefficients of $r \geq 0.95$, indicating that the LP aerosol data are reliable in that altitude range. Comparisons of extinction time series show that both instruments capture the variability of the stratospheric aerosol layer quite well, and the differences between the two instruments vary from 0% to ±25% depending on altitude, latitude, and time. On the other hand, we find erroneous seasonal variations in the OMPS/LP Version 1.5 dataset, which usually exist below 20km in the Southern Hemisphere due to the lack of sensitivity to particles when scattering angle (SA) is

greater than 145°. We also find that LP retrieved extinction is systematically higher than SAGE III/ISS observations at altitudes above 28 km and systematically lower below 19 km in the tropics with significant biases up to ±13%. This is likely due in part to the fact that the actual aerosol size distribution is altitude dependent. There are other reasons related to cloud contamination, wavelength limitations, aerosol loading and the influence of the viewing configuration.

## 1. Introduction

The stratospheric aerosol layer is an important component of Earth's atmosphere through its impacts on climate and stratospheric ozone physico-chemistry (Vernier et al., 2011; Ridley et al., 2014; Bingen et al., 2017). The stratospheric aerosol layer was first observed by Junge in 1960 (Junge et al., 1961). Stratospheric aerosols that mainly originate from volcanic sources are described as liquid droplets composed of a mixture of the sulfuric acid ($H_2SO_4$) and water ($H_2O$) (Kremser et al., 2016). Recent measurements show that the background stratospheric aerosol layer is variable rather than constant, and that changes in the stratospheric aerosol layer have caused observable changes in recent tropospheric warming rates (Solomon et al., 2011), indicating that it is important to monitor the stratospheric aerosol layer over the long term. The importance of possible changes in the background stratospheric aerosol layer led to the analysis of each volcanically quiescent period (Deshler et al., 2006). For the considered period from January 1979 through the end of 2004, the variability of stratospheric aerosol layer is explored with measurements from space-based instruments such as SAGE II (Thomason et al., 2008), CALIPSO (Winker et al., 2010), GOMOS/ENVISAT (Vanhellemont et al., 2010), SCIAMACHY (von Savigny et al., 2015), OSIRIS/Odin (Bourassa et al., 2007), SAGE III/ISS (Chu et al., 1998) and OMPS/LP (Loughman et al., 2018).

The OMPS Limb Profiler (LP) is one of three OMPS instruments onboard the Suomi National Polar-orbiting Partnership (S-NPP) satellite (Flynn et al., 2007). S-NPP was launched in October 2011, into a sun-synchronous polar orbit with 13:30 local equator crossing time. The LP instrument collects limb scattered radiance data on a 2-D charge coupled device (CCD) array over a wide spectral range (290-1000 nm) and a wide vertical extent (0–80 km) through three parallel vertical slits. These spectra are primarily used to retrieve vertical profiles of ozone (Rault and Loughman, 2013; Kramarova et al., 2018), aerosol extinction coefficient (Loughman at al.,

2018; Chen et al., 2018), and cloud-top height (Chen et al., 2016).  The vertical sampling of LP measurements is about 1 km, and the vertical resolution of the retrieved profiles is approximately 1.6 km. More details about the OMPS/LP instrument design and capabilities are provided in Jaross et al. (2014).

Recently, a new aerosol size distribution (ASD) based on a gamma function that is derived from Community Aerosol and Radiation Model for Atmospheres (CARMA) calculated results was introduced in the Version 1.5 OMPS/LP aerosol extinction retrieval algorithm (Chen et al. 2018). The assumed ASD is designed to represent the long-term background stratospheric aerosol loading. We use the CARMA model output to take advantage of its large range of

simulated particle size information, and we find that a gamma size distribution represents a significantly better fit to its size distribution than a lognormal distribution. In a recent study, Nyaku et al. (2019) confirms that the CARMA model agrees well with the Wyoming optical particle counter (OPC) measurements. Chen et al. (2018) tested the Version 1.5 algorithm by comparing 7 months of data (June -December 2017) between OMPS/LP and the Stratospheric

Aerosol and Gas Experiment III instrument onboard the International Space Station (SAGE III/ISS). The limited comparison showed the consistency of the aerosol extinction measurements from both instruments, with a correlation coefficient of 0.97 and slope  of 1.05. The Version 1.5 OMPS/LP aerosol products are now being processed routinely, and additional SAGE III/ISS data are available.

This work extends the previous results shown in Chen et al. (2018) to evaluate LP aerosol extinction profiles through comparison with independent data sets from SAGE III/ISS. The central scope of the paper is the evaluation of the LP algorithm performance for background aerosol situations. We document here a more comprehensive evaluation of this new version of the OMPS/LP aerosol product. The latest versions, i.e., Version 1.5 of LP and Version 5.1 of

SAGE III/ISS, are used. The objective of this comparison is to assess the reliability of the LP Version 1.5 algorithm and to identify potential problems. The differences of aerosol extinction values observed by the two instruments are analyzed and discussed. Our analysis of SAGE III/ISS data specifically addresses possible biases with OMPS/LP results arising from differences in vertical resolution and possible ozone contamination. The impact of volcanic perturbations on

the retrievals is also investigated.

## 2. LP Algorithm Description

The previous version 1 aerosol retrieval algorithm for OMPS/LP is described in detail by Loughman et al. (2018). Here, we provide a brief description of key changes implemented in the retrieval algorithm for processing the V1.5 dataset.

In the V1.5 algorithm, a gamma function based ASD is assumed (Chen et al., 2018):

$$n(r) = \frac{dN}{dr} = \frac{N_0 \beta^\alpha r^{\alpha-1}}{\Gamma(\alpha)} \exp(-r\beta) \qquad (1)$$

where $n(r)$ is the number of particles $N$ per unit volume with a size between radius $r$ and $r+dr$ ($cm^{-3}\mu m^{-1}$), $N_0$ is the total number density of aerosols ($cm^{-3}$), $\Gamma$ is Euler's gamma function, $\alpha$ is the shape parameter and $\beta$ ($\mu m^{-1}$) is the scale parameter. At small radii this function follows a power law, while at large radii it follows an exponential function. The cross-section and aerosol scattering phase function (PF) are then calculated using Mie theory assuming liquid sulfate spherical particles with a refractive index from Russell et al. (1996). The $\alpha$ and $\beta$ parameters were determined by fitting the gamma distribution to size distributions for stratospheric sulfate simulated by the Community Aerosol and Radiation Model for Atmospheres (CARMA) module running online in the Goddard Earth Observing System (GEOS) global model (after English et al. 2011, Colarco et al. 2014). The fitted ASD yields an Angstrom Exponent (AE) of 2.08 and an effective radius ($r_{eff}$) of 0.18 $\mu$m, similar to the average values determined from SAGE II version 7.0 aerosol extinction data (Thomason et al., 2008; Damadeo et al., 2013) between 20 and 25 km (red and green dots in Figure 1), from measurements at 525 and 1020 nm taken during the period 2000-2005, which is characterized by a low volcanic background. This was a period when the stratosphere was relatively clean and roughly similar to the present day stratosphere. Note that average SAGE II AE values at 30 km (blue dots in Figure 1) are larger than that at lower altitudes. Other time periods known to have low aerosol loading (e.g. 1989-1990) show lower values of AE between 20 and 25 km in the SAGE II dataset. Thus, the reference ASD adopted here for LP retrievals may produce a bias in extinction values derived during medium or high volcanic aerosol loading. The SAGE II observed variability of aerosol properties with altitude is related to variability in temperature and humidity fields that affects the aerosol size distribution and refractive index (Steele and Hamill, 1981; Russell et al, 1996), both of which have a direct effect on the PF. This variability also and more significantly depends on

the aerosol load, in the stratosphere, especially if a large time period is considered as in Figure 1. Assuming that the aerosol signal in line of sight radiances is roughly proportional to the PF, aerosol extinction profiles at 675 nm are retrieved using an iterative technique (see Sect. 4.2 of Loughman et al., 2018), based on Chahine's non-linear relaxation technique (Chahine, 1970). Atmospheric pressure and temperature profiles used in this retrieval algorithm are obtained from the GEOS atmospheric analyses produced by NASA GSFC Global Modeling Assimilation Office (GMAO). For the LP aerosol product, retrievals are only performed for daytime observations (solar zenith angle SZA < 88°).

## 3. Evaluation Analysis
### 3.1 SAGE III/ISS data

The SAGE III/ISS developed by the NASA Langley Research Center (LaRC) was launched to the International Space Station in February of 2017. SAGE III/ISS makes sunrise and sunset occultation measurements of aerosols and gas concentrations in the stratosphere and upper troposphere (Chu et al., 1998). The ISS travels in a Low-Earth orbit at an altitude of 330-435 km and at an inclination of 51.6°. With these orbital parameters, solar occultation measurement opportunities cover a large range of latitudes (between 70°S and 70°N). The instrument measures up to 31 combined sunrise and sunset profiles each day. A general description of the solar occultation measurement technique is provided by McCormick et al. (1979). Aerosol extinction at nine wavelengths (384.2, 448.5, 520.5, 601.6, 676.0, 756.0, 869.2, 1021.2 and 1544.0 nm) are provided by SAGE III/ISS from the surface or cloud top to an altitude of 45 km, with a vertical resolution of 0.5 km at the tangent point location. The SAGE III/ISS series of aerosol occultation measurements have been extensively evaluated and compared with other space based instruments and have been found to have relatively high precision and accuracy (Thomason et al., 2010, 2018; Bourassa et al., 2012; Kovilakam and Deshler, 2015; von Savigny et al., 2015; Rieger et al., 2018). In this work we use SAGE III/ISS Version 5.1 data, including both sunrise and sunsets, which were collected during the period June 2017 – May 2019. Figure 2 shows the spatial and temporal coverage of the available datasets during the 2017 to 2019 period considered in this study.

### 3.2 Methodology

Chen et al. (2018) described previous LP-SAGE III/ISS comparisons using SAGE III/ISS Version 5.0 data for the period of seven months from June to December of 2017. The first comparison was conducted in two steps. First, all data were binned and averaged in 10° latitude bins for each altitude for groups of 1-3 consecutive days depending on the number of SAGE III/ISS samples. In the second step, the averaged extinction profiles at 675 nm for LP and at 676 nm for SAGE III/ISS were compared directly. In this work, both data sets were averaged zonally for each 5° latitude band per day at each of their respective altitudes. The daily averaged data in the same latitude bin and on the same day were used for the comparison with an assumption that the variation with longitudinal bands is much smaller than the variation with latitudinal bands. For background aerosol situations, the longitudinal variation of stratospheric aerosol could be small because of efficient mixing in the zonal direction and strong horizontal transport prevailing in the region (Sunilkumar et al., 2011). In cases of medium to large volcanic eruptions, however, longitudinal differences could be large and a restriction should be placed on longitude to capture the signature of potential longitudinal variation. Only LP data from the center slit were taken into consideration because the center slit has better straylight and tangent height corrections compared to the left and right slits (Moy et al., 2017; Kramarova et al., 2018). The current cloud detection algorithm (Chen et al., 2016) detects cloud top height from the OMPS/LP measurements using the spectral dependence of the vertical gradient of radiances at 675 and 868 nm. Cloud top height is identified when the gradient difference increases above 0.15. All LP data below the cloud top height were rejected because extinction changes abruptly at cloud top.

As the SAGE III/ISS science team pointed out (Thomason, L., personal communication, 2019; Wang et al., 2020, submitted to JGR), reported SAGE III/ISS aerosol extinction is retrieved as a residual of using a spectrally-focused fitting (i.e., it derives from the "MLR" ozone product) and any bias in ozone would result in a bias in the aerosol extinction values in the vicinity of the Chappuis bands depending upon altitude and ozone concentration. To avoid possible biases in the SAGE III/ISS reported 676 nm aerosol extinction, associated with remaining ozone absorption effects, the aerosol channels at 449 nm and 756 nm were used in this study to interpolate to 675 nm SAGE III/ISS extinction profiles using a log-linear interpolation. Figure 3a shows SAGE III/ISS profiles of reported (black line) and recalculated by spectral interpolation (green line) 675 nm aerosol extinction at 0.5 km vertical resolution. Large differences between reported and recalculated SAGE III/ISS extinction values are apparent

above 27 km. The extinction minimum at 29 km in the original data is not present in the interpolated profile. Also shown is the corresponding LP profile (blue line) at 1.0 km sampling grid. Since the SAGE III/ISS data are given at 0.5 km intervals, while the vertical resolution of LP is around 1.6 km, the vertical resolution of SAGE III/ISS was then degraded to match the OMPS/LP vertical resolution using a 7-point binomial smoothing function given by the expression:

$$k(z) = \sum_{i=1}^{7} k(z + \Delta z_i) * w(i);$$
$$\Delta z_i = -1.5, -1, -0.5, 0.0, 0.5, 1, 1.5; \qquad (2)$$
$$w(i) = [1, 6, 15, 20, 15, 6, 1] / 64$$

where $k$ is the extinction, $w$ is the weighing factor and $z$ is the altitude grid point. This procedure approximates a Gaussian smoothing of continuous data with full width at half maximum of about 1.6 km. This spectrally interpolated and vertically smoothed SAGE III/ISS data is used for all comparisons in this paper. Figure 3b provides an example of comparison of extinction profiles between the LP retrieval profile (blue line) and the smoothed version of the recalculated SAGE profiles (red line). The figure shows how LP and the smoothed+interpolated SAGE III/ISS profile are similar above 19.5 km, and have more disagreement below this altitude. In the upper troposphere and lower stratosphere (UTLS), particularly in the tropics, the presence of thicker aerosol layers may have an impact on the retrieved extinction levels (Bourassa et al.,2012; Fromm et al., 2014; Kremser et al. 2016). In addition, sea salt can also increase the particle size in the UTLS due to water uptake (Brühl et al., 2018). Cirrus cloud contamination could be another issue in the lower stratosphere below about 19 km. Clouds appear as discontinuities of the limb radiance vertical profiles, which will tend to bias the retrieved result. Although most clouds are detected and filtered from the LP retrievals, it is not always possible to completely eliminate cloud contamination. Furthermore, large differences (about 60%) between the two instruments are expected at lower altitudes near and below the tropopause because of larger variability in the transport of air masses.

## 4. Results and discussion

### 4.1 Aerosol Extinction Variability

Figure 4 depicts the time series of Version 1.5 OMPS/LP (blue symbols) and SAGE III/ISS (red symbols) at 20.5 km, 25.5 km and 30.5 km in the tropics from June 2017

through May 2019. The time dependent variability of each dataset is similar and data features associated with specific aerosol events observed by SAGE III/ISS are also present in the LP record. The observed large aerosol extinction increase at 20 km apparent in both data records by the end of 2018, is associated with the eruption of Mt. Ambae (June 2018 at 15°S). For the background aerosol condition based on the June, 2017 to June 2018 period, LP and SAGE III/ISS are within 20% of each other at 20.5 km and 25.5 km, indicating that the assumed ASD in LP V1.5 algorithm is adequate in this altitude range. The large temporal variability present in both datasets at 30.5 km is suggestive of a quasi-biennial oscillation (QBO) signal, although there is a shift in phase between latitude bands. At 30.5 km, the agreement is not quite as good, LP retrievals are systematic higher than SAGE III/ISS extinctions for all latitude bins with positive biases up to 50%. The large biases reflect the very small aerosol load at altitudes above 30 km. Under low aerosol condition, both instruments are less sensitive to smaller particles (Rieger et al., 2018). For example, the AE values from SAGE II are quite scattered at 30 km compared to lower altitudes (see Figure 1), which may be related to reduced quality of the aerosol retrieval for low aerosol loading. On the other hand, the systematic positive bias between LP and SAGE III/ISS may be due in part to the fact that the actual ASD, and refractive index, which depend on temperature and humidity (Steele and Hamill, 1981; Russell et al, 1996), are not truly independent of height as currently assumed in the LP algorithm. While the SAGE III/ISS algorithm does not require any assumptions about aerosol microphysics, aerosol extinction profiles from OMPS/LP suffer from uncertainties due to assumed ASD and refractive index. Additionally, uncertainties in LP radiance measurements are assumed to be 1 % (Kramarova et al., 2018) and the primary source of error in LP radiance measurements is the straylight error which increases with altitude. At higher altitudes where LP radiance is small, the straylight error becomes most significant.

Figure 5 shows the time series comparison between OMPS/LP and SAGE III/ISS measurements at 15.5 km (left panel), 20.5 km (middle panel) and 25.5 km (right panel) for different latitude bands outside the tropics where clouds are not an issue. The blue and black dots in Figure 5 represent the LP extinction calculated at scattering angles (SA) < 145° and > 145°, respectively. Again, the highly variable nature of the stratospheric aerosols with time and latitude is well represented by the two instruments. Canadian pyro-cumulonimbus (PyroCb) was most probably responsible for increasing aerosol extinction values at 15.5 and 20.5 km in the 35°N-

55°N latitude bands in late 2017, and the effect of Mt. Ambae can be seen in the 35°S –55 °S time series in late 2018. Good agreement is found between both instruments at 20.5 and 25.5 km, although there are some negative biases in the SH. In contrast to the results for altitude at 30.5 km (right panel in Figure 4), the LP values at 15.5 km are systematically smaller than the ones from SAGE III/ISS. A discussion of this systematic difference is given in Section 4.3.

A notable feature in Figure 5 is that the comparisons in the Northern Hemisphere (NH) are generally better than in Southern Hemisphere (SH). In the southern mid-latitudes (35°S-55°S), the LP retrievals show significant seasonal variations at 20.5 and 25.5 km that are not seen by SAGE III/ISS. The obvious seasonal variability in the differences, with the amplitudes of winter minima was observed to be about 25% at 20.5 km, and as much as 200% at 15.5 km. The presence of erroneous seasonal variations in the OMPS/LP dataset is mostly caused by limitation of wavelength at 675 nm when observing in backscatter condition at extreme large scattering angles. The results lead us to recommend filtering LP data below 20 km with SA greater than 145°. The limitations of the LP retrievals at 675 nm will be addressed in Section 4.5.

## 4.2  Effect of volcanic eruptions and PyroCbs

To investigate the impacts of volcanic eruptions and intense wildfires on the LP retrievals, aerosol extinction profiles from OMPS/LP and SAGE III/ISS were inter-compared in the aftermath of the eruption of Mt. Ambae (2018) and the Canadian wildfires (2017) sampled by these instruments.

Volcanic eruptions are the largest source of long-lived aerosols in the stratosphere (Vernier et al., 2011; Kremser et al., 2016; Bingen et al., 2017). The Ambae eruption occurred on July 27, 2018, in Ambae Island, located near 15° S, 167° E, and had a clear impact on the stratospheric aerosol load. The top panel of Figure 6 shows comparisons of LP and SAGE III zonally averaged stratospheric aerosol profiles between 0–5°N on three days before the eruption (first three plots), along with the relative differences on each day (fourth plot). Pre-eruption LP measurements agree within 5% of SAGE III/ISS observations over the 20-25 km altitude range. SAGE III/ISS measurements were not available at 0–5°N until approximately three weeks after the eruption, due to the complex sampling pattern shown in Figure 2. OMPS/LP data show significant longitudinal transport of the stratospheric aerosol plume during this period, so we have used zonal mean averages to compare post-eruption results as well. Larger differences (up

to 20%) are observed for post-eruption conditions as shown on a similar set of plots shown on the bottom panel of Figure 6. The enhanced extinction around 20 km after the eruption was captured by both OMPS/LP and SAGE III/ISS observations. Much larger differences between the two data sets are apparent below 20 km and above 25 km both before and after the volcanic eruption.

In addition to volcanic eruptions, intense wildfires can also cause aerosol perturbations in the UTLS. The occurrence of PyroCbs triggered by intense wildfires in British Columbia, Canada were reported on August 12, 2017. Within two months after injection the plume can reach up to nearly 22 km (Peterson et al., 2018; Torres et al. 2020) and was transported globally (Kloss et al., 2019). A before-and-after analysis, was carried out to evaluate the performance of the LP algorithm following the stratospheric injection of carbonaceous aerosols. Zonal averages of SAGE III/ISS and LP aerosol extinction retrievals for the 45°N–50°N latitude band before and after the aerosol injection effect were compared. Figure 7 highlights the emergence of new stratospheric aerosol layers before and after the fire event in Canada in late 2017. Both instruments clearly captured the stratospheric aerosol perturbation triggered by the reported PyroCb. We chose July 7, 2017 as the "before" case (Figure 7a). SAGE III/ISS observations at high northern latitudes were not available until September 2017, during which time the enhanced stratospheric aerosol spread rapidly in longitude. SAGE III/ISS results in September-October 2017 at 50°N show significantly increased scatter. We therefore calculated the difference for LP and SAGE III/ISS separately using November 10, 2017 as the "after" case (Figure 7b) to show the anomaly results. As expected, Figure 7c shows large positive extinction anomalies of up to 140 % below 22 km. The comparison also shows good agreement between LP retrieved extinction profiles and observations from SAGE III/ISS within ±20 % at 18-28 km. Since the LP algorithm assumes a fixed refractivity index which is dependent on aerosol type, the wrong aerosol type (sulfate instead of carbonaceous aerosols) leads to an incorrect phase function, which then produces an error in the retrieved extinction. For SAGE III/ISS, this assumption is not needed, so there is less impact.

**4.3 LP retrievals above 27 km and below 19 km**

All comparisons shown in Figures 4–7 have some features in common. The best agreement is found for the altitude range roughly between 19 and 27 km, but systematic and

significant differences are observed above 28 km and below 19 km. These systematic differences change with altitude and may be associated with the algorithm assumption of time and altitude independent aerosol model (i.e., refractive index and ASD) and the corresponding PF. SAGE III/ISS can't measure the PF but the aerosol spectrum (in this case the approximation of the AE) can be used as an effective qualitative indicator of particle size (Hayashida et al., 2001; Schuster et al., 2006; Damadeo et al., 2013; Rieger et al., 2018).

To examine the LP algorithm performance of the aerosol retrieval due to the assumed ASD, we perturbed the nominal gamma ASD fitting parameters so that the resulting AE yielded values of 2.3 and 1.8 (about ±10% of the algorithm's baseline value of 2.08). The 2.3 value is close to the reported average SAGE II AE at 30 km for the period 2000-2005 (blue dots in Figure 1) whereas the AE value of 1.8 resembles similar SAGE II average at 18 km in the case of reduced aerosol load. The perturbed ASD's, associated with AE values of 1.8 and 2.3, were used to estimate the resulting effect on the calculated 675 nm PF as shown in Figure 8. A ±10% change in AE can produce a about ±15% change in PF. The results of the sensitivity analysis in Figure 8, indicates that changes in assumed PF will result in significant changes in retrieved aerosol extinction values.

Figure 9 shows that the use of the baseline AE value (2.08) produces significant error (up to ±13%) in retrieved extinction at 30.5 km and 18.5 km relative to the outcome when using AE values comparable to those from SAGE II v7.0 data. The results of this sensitivity analysis show that when taking into account the altitude dependency of stratospheric aerosol properties, the LP algorithm performance improves at those levels. Figure 10 shows differences in aerosol extinction between OMPS/LP and SAGE III/ISS at 30.5 km in the tropics (top) and at 18.5 km in northern mid-latitudes (bottom) plotted as a function of LP scattering angle (SA) for the entire comparison period to examine if the assumed ASD is reasonable. As discussed earlier, if the assumed ASD is correct, the difference in extinction should be SA independent in the SA range 50°–100°. The differences shown in Figure 10 vary with SA in that range significantly, suggesting the currently assumed ASD in the LP algorithm does not adequately represent the actual ASD for altitudes at 30.5 km and 18.5 km. As a result, the potential ASD error propagates into extinction uncertainties above 28 km and below 19 km. Another important aspect that influences the comparison below 19 km is the presence of clouds and thicker aerosol layers

discussed earlier (Figure 3). Near or below the tropopause, big disagreements between the two instruments can also be expected due to the variability of the transport of air masses.

The sensitivities given in Figures 8-10 suggest that the differences in extinctions between the two instruments could be partly explained with the variation of the AE. Figure 11 illustrates how the AE varies. In the figure, monthly mean AE (left panel) derived from the vertically smoothed SAGE III/ISS data calculated using Eq.2 and differences between SAGE III/ISS and OMPS/LP extinction values at 675 nm (right panel) are plotted as a function of altitude and time at different latitude bands in the tropics. To draw a corollary to SAGE II, the AE is derived using the aerosol extinction values at 520 nm and 1020 nm. However, to avoid potential bias, the aerosol extinction is first interpolated to 520 nm using a second order polynomial in log-log space (extinction versus wavelength) with extinction values at 448nm, 756 nm, 869 nm, and 1020 nm. The computed AE is plotted on a color scale with the baseline value of 2.08 used for the OMPS/LP algorithm at its center. Red colors in Figure 11a thus represent SAGE III/ISS AE values larger than the OMPS/LP AE value, while blue colors represent AE values smaller than the OMPS/LP value. AE values above the baseline are possibly associated with the volcanic eruption of Mt. Ambae below 25 km (note transport from the QBO in the lower center of the panels in Figure 11a) and aerosol evaporation at higher altitudes, particularly in the tropics (Schuster et al., 2006; Kremser et al., 2016; Rieger et al., 2018; Malinina et al., 2019). AE values are typically below the baseline through the mid-stratosphere in non-volcanic conditions, though the smallest values in the upper troposphere are possibly associated with clouds (and a minor influence from the PyroCb in the lower stratosphere at the northern extent of the plotted data in late 2017). SAGE III/ISS data suggest that the overall AE throughout most of the stratosphere is slightly below the baseline value of 2.08 used by the OMPS/LP algorithm. The similar structure between AE variations shown in Figure 11a and the extinction differences shown in Figure 11b suggest that the use of an altitude-dependent ASD in the LP retrieval is a significant component of the observed extinction differences, although other factors may also contribute. Section 4.5 gives further discussion of how the assumed LP phase function and LP measurement geometry can result in latitude-dependent variations in extinction differences with SAGE III/ISS.

**4.4 Regression analysis**

Figure 12 shows scatter plots of individual daily zonal mean extinction values from OMPS/LP and SAGE III/ISS for 5° latitude bands for 60° S -30° S (left panel), 30° S - 30° N (middle panel) and 30° N - 60° N (right panel). The corresponding comparison statistics are written within the plot. While the overall impression from these results is encouraging, the agreement changes with latitude. In the tropics and northern mid-latitudes (30° S - 30° N and 30° N - 60° N), there is a good agreement between the results from both instruments, with most observations close to the 1:1 line and a correlation coefficient greater than 0.96. This result gives further quantitative evidence that the assumed ASD is appropriate for OMPS/LP aerosol extinction retrievals in most of the stratosphere for background aerosol. In the southern mid-latitudes between 60° S -30° S, zonal means from LP and SAGE III/ISS are in fair agreement with a correlation coefficient of 0.95, but large systematic differences (SAGE III/ISS greater than LP) are observed for aerosol extinction, $k > 0.0001$ m$^{-1}$. These positive deviations up to 25% could be related to wavelength limitations (see the next section).

**4.5 Wavelength Dependence  on OMPS/LP Aerosol Sensitivity**

In Figures 5 and 12, the comparison shows asymmetry between the hemispheres below 20.5 km, with much better agreement in the NH than in the SH, and OMPS/LP extinction values are significantly biased at southern mid-latitudes below 20 km due to erroneous seasonal variations in the OMPS/LP dataset. That suggests the LP measurements are more sensitive to aerosols in the NH and less sensitive to those in the SH, especially at lower altitudes. Here we shall examine the sensitivity of the LP radiances to aerosol.

As mentioned before, the LP V1.5 algorithm uses OMPS/LP radiances at a single wavelength at 675 nm to retrieve extinction profile. This wavelength was selected primarily to minimize aerosol-related errors in the ozone retrieval and to reduce stray light contamination (Loughman et al., 2018). However, as indicated in Sections 4.1 and 4.4, it is difficult to retrieve reliable aerosol extinction below 20 km in the SH due to lack sensitivity to aerosol. Figure 13 shows an example of aerosol weighting functions at 675 nm for three latitudes. The aerosol weighting function, which determines how the calculated radiance (*I*) at a given wavelength changes with a change in aerosol extinction (*k*), is denoted by:

$$\frac{\partial \ln(I)}{\partial \ln(k)} \qquad (3)$$

The derivatives are calculated for all altitudes for a change at each tangent height, and each curve in the figure shows the sensitivity of the radiance at a given tangent height to extinction perturbations of a 1 km layer at a range of altitudes. It can be seen that the sensitivity to aerosol varies with latitude and altitude. The LP radiance at 675 nm is most sensitive to the aerosol extinction over the 20-30 km altitude range in the tropics and the northern mid-latitudes, but less sensitive to aerosol in the southern mid-latitudes. Therefore, uncertainties in OMPS/LP aerosol retrievals at 675 nm increase with reduced sensitivity to aerosol extinction because lower sensitivities to aerosol may result in noise amplification. This behavior is consistent with the results shown in Figures 5 and 12 that the LP retrievals are in better agreement with the SAGE III/ISS data in the NH than those in the SH.

In the limb scattering technique, on the other hand, the aerosol signal in the limb radiance at a given tangent height is roughly proportional to the product of the aerosol extinction in that layer and the PF at the tangent point. OMPS/LP is installed in a fixed orientation relative to the S-NPP spacecraft. Therefore, southern mid-latitudes are observed at backscattering geometries whereas NH observations are carried under forward scattering conditions (see Loughman et al., 2018, Figure 2). Aerosol scattering phase function is at least fifty times smaller for OMPS limb viewing in SH than in NH. Due to the variation of the PF with latitude and season, the LP observations are most sensitive to aerosols in the NH winter and least sensitive to those in the SH. LP scattering angles typically vary between 15 and 165°. For the selected wavelength of 675 nm and the assumed ASD, the PF has much smaller values at larger scattering angles (see Chen et al., 2018, blue line in Figure 2), corresponding to the larger solar azimuth angles at southern latitudes. This leads to a smaller relative contribution of aerosol scattering with respect to Rayleigh scattering. At extreme large scattering angle (>145°) where Rayleigh scattering is high and the value of the aerosol phase function is close to its minimum, the LP radiances at 675 nm are nearly insensitive to aerosol at lower altitudes. Therefore, the SA upper limit (= 145°) should be used to filter the data. Outside of this SA range, LP extinction values will tend to bias the retrieved result, as revealed in Figure 5. This could explain the presence of erroneous seasonal variations in the OMPS/LP dataset.

Since the sensitivity to aerosol increases at longer wavelengths (Taha et al., 2011), the uncertainties in the southern mid-latitudes at lower altitude may be reduced by using longer wavelengths. Figure 14 illustrates the sensitivity of the limb radiances to the aerosol extinction at

three wavelengths for latitude at 60 °S (solar zenith angle SZA = 70°). The sensitivity to aerosol is seen to increase with increasing wavelength. The increased sensitivity of the limb radiances to aerosol at longer wavelengths is partly due to the fact that the Mie scattering from aerosol particles does not decrease as rapidly with wavelength as the Rayleigh scattering from air
molecules (Bourassa et al., 2007).

## 5. Summary and Conclusions

To extend our previous validation of the OMPS/LP aerosol products, OMPS/LP Version 1.5 extinction profiles have been compared against SAGE III/ISS Version 5.1 aerosol data over
the period June 2017 to May 2019. For the comparison, both LP and SAGE III/ISS extinction profiles in this paper are in the form of daily averages for a 5° latitude bin at each altitude. To evaluate the 675 nm extinction profiles from OMPS/LP, the original SAGE III/ISS extinction values at 449 nm and 756 nm are used to interpolate to the extinction profiles at 675 nm, to avoid contamination from potential biases in the SAGE III/ISS "MLR" ozone product, and then
degraded vertically to the same 1.6 km vertical resolution as OMPS/LP. Using SAGE III/ISS aerosol channels away from the Chappuis bands and matching the vertical resolution of the instruments significantly improves the comparisons.

Overall, results show very good agreement for extinction profiles between 19 to 27 km, to within ±25%, and show systematic differences (LP-SAGE III/ISS) above 28 km and below 19
20   km (> ±25%). The results show that the LP retrievals are in better agreement with the SAGE III/ISS data in the NH than those in the SH, and the agreements change with time and altitude. Comparisons of time series show that both instruments detect changes in stratospheric aerosol layer well during the entire time period of June 2017~ May 2019. To investigate the impact of volcanic perturbations on the retrievals, the differences of extinction profiles measured by LP
and SAGE III/ISS in the same latitude bin and on the same day are analyzed before and after the occurrence of the Mt. Ambae volcanic eruption and the Canadian PyroCb. The results show that both instruments can capture the volcanic aerosol and PyroCb well, and their discrepancies are small (< ±10%), indicating that the LP retrieved extinction profiles are reliable under these conditions. In order to better quantify the difference in the variability observed in the OMPS/LP
and SAGE III/ISS measurements, a total of 9862 data points between 19.5 km and 27.5 km were used to perform a linear regression analysis. The regression results show very high correlations

with the linear correlation coefficient greater than 0.95, and most observations lie close to the 1:1 line, except for the southern mid-latitudes where large positive differences up to 25% are observed due to differences in retrieval techniques and sensitivity issue. These comparisons are consistent with the previous analysis performed by Chen et al. (2018) that demonstrated a good agreement between the extinction profiles in the range of 19 to 29 km, and much larger differences above 30 km and below 19 km.

At altitudes above 28 km (below 19 km), LP extinction values were systematically higher (lower) than SAGE III/ISS observations. These significant and systematic biases between LP and SAGE III/ISS are partly due in part to the fact that the actual ASD is both altitude and time dependent. To evaluate the LP performance of the aerosol retrieval from the assumed ASD, the averaged Angstrom exponent (AE) derived from SAGE II v7.0 data under background aerosol conditions was compared with the baseline AE derived from the assumed ASD. The analysis indicates that in background conditions, the baseline AE is underestimated (overestimated) above 28 km (below 19 km), thus the assumed ASD provides a less accurate representation at altitudes above 28 km (below 19 km), tending to overestimate (underestimate) extinction values above 28 km (below 19 km). Our sensitivity analysis shows that the ASD error propagates into extinction uncertainty as much as 13%. This suggests that a dynamic model for ASD is needed to accurately retrieve aerosol extinction profiles. There are three other possible causes for the discrepancies. First, most of the remaining discrepancies in the comparison are possibly attributed to the inherent uncertainties associated with the measurement techniques. In principle, the differences in the measurement techniques affect the instrument sensitivity to aerosols (Malinina et al., 2019). While SAGE III/ISS uses the solar occultation measuring technique which is self-calibrating and derives extinction directly, OMPS/LP employs the limb scatter technique where retrieved extinction depends on instrument calibration and tangent height registration as well as the inversion algorithm and its several assumptions including the assumed aerosol microphysics. Uncertainties in these assumptions can also affect the LP extinction product. Second, some discrepancies at higher altitudes could be also caused by reduced aerosol loading and hence reduced retrieval accuracy for both instruments. The limb radiance is also more susceptible to additive straylight contamination at high altitudes, where absolute radiance values are smaller. Third, large differences between LP and SAGE III/ISS at lower altitudes of as

much as 60% may also be associated with thin cirrus clouds, thick aerosol layer and air mass transport at various locations and levels.

Another significant finding in this study is that the comparison shows asymmetry between the hemispheres below 20.5 km, with better agreement in the NH than in the SH, and erroneous seasonal variations at southern latitudes in the OMPS/LP dataset with an estimated uncertainty of a factor up to two. The reason for this is identified as that the LP radiances at 675 nm are nearly insensitive to aerosol at extremely large scattering angles. This problem can be solved by using longer wavelengths to retrieve aerosol extinction. In the meantime, we recommend filtering LP data below 20 km with scattering angle greater than 145°. When scattering angle exceeds this limit the LP algorithm starts to give obviously wrong results.

OMPS/LP Version 1.5 aerosol profiles are freely available to the international research community at https://disc.gdfc.nasa.gov.

**Acknowledgements**

We thank the OMPS/LP team at NASA Goddard and Science Systems and Applications, Inc. (SSAI) for help in producing the data used in this study. We also thank Larry Thomason at the NASA Langley Research Center for his insights into the stratospheric aerosol problem, and Philippe Xu and Tong Zhu for their technical support. SAGE III/ISS is a NASA Langley managed Mission funded by the NASA Science Mission Directorate within the Earth Systematic Mission Program. Enabling partners are the NASA Human Exploration and Operations Mission Directorate, International Space Station Program and the European Space Agency.

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

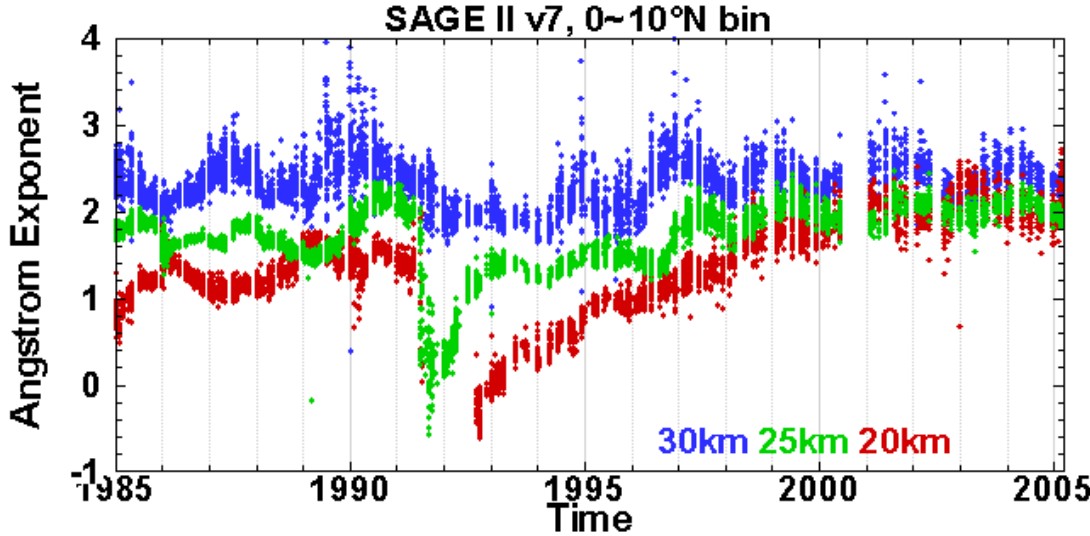

**Figure 1.** Time series of SAGE II Angstrom Exponent (AE) derived from the aerosol extinction coefficients measured at 525 nm and 1020 nm for altitudes of 30 km (blue), 25 km (green) and 20 km (red). This figure shows SAGE II version 7.0 data for the 0–10° N latitude band during the period 1986 - 2005. While the Pinatubo eruption in 1991 produced a significant decrease in AE, the smaller volcanic eruptions such as Ruang/Reventador in 2002 and Manam in 2005, visible in the extinction time series (not shown), did not appreciably affect AE values. The AE values appear to stabilize after 2000, suggesting that a background state exists. The AE is quite scattered at 30km compared to at lower altitudes, which is related to reduced quality of the aerosol retrieval for low aerosol loading.

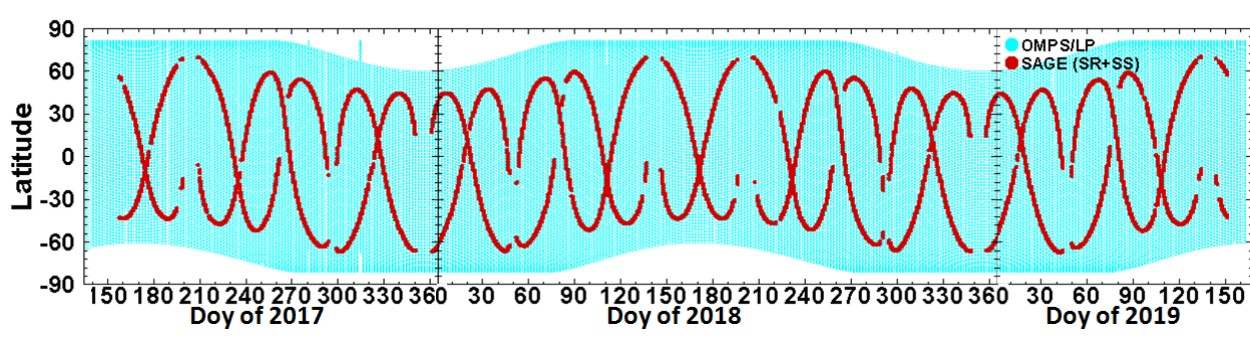

**Figure 2.** Time-latitude sampling pattern for OMPS/LP (light blue) and SAGE III/ISS (red) during the period June 2017 – May 2019.

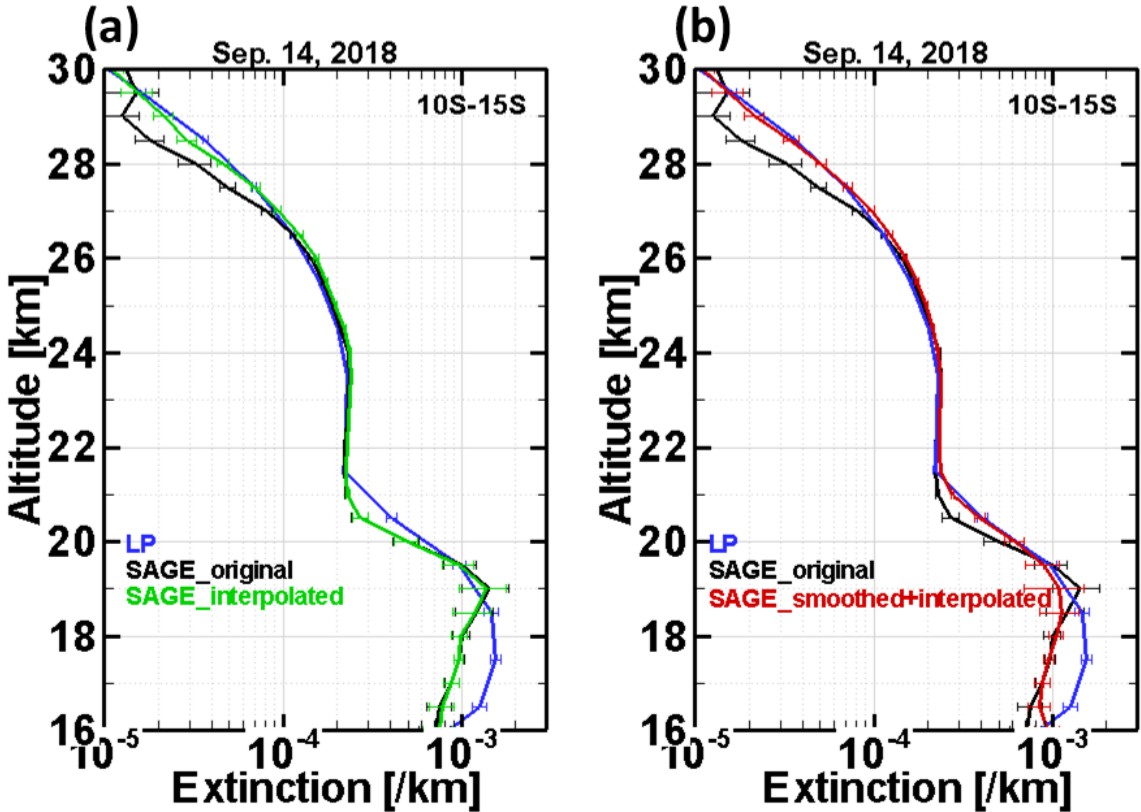

**Figure 3.** An example LP-SAGE comparison of average extinction profiles in the latitude band 10–15°S
for one day, September 14, 2018. **(a)** Comparison of extinction profiles among original SAGE (black),
interpolated SAGE (green) and retrieved LP (blue). **(b)** Comparison of extinction profiles between the
smoothed version of the recalculated SAGE profile (red) and the LP retrieval profile. Applying a
combination of the smoothing and interpolation approaches to the original SAGE data improves the
comparison above 19 km. The horizontal error bars on the mean extinction profiles indicate standard error
of the mean, $\sigma/\sqrt{N}$.

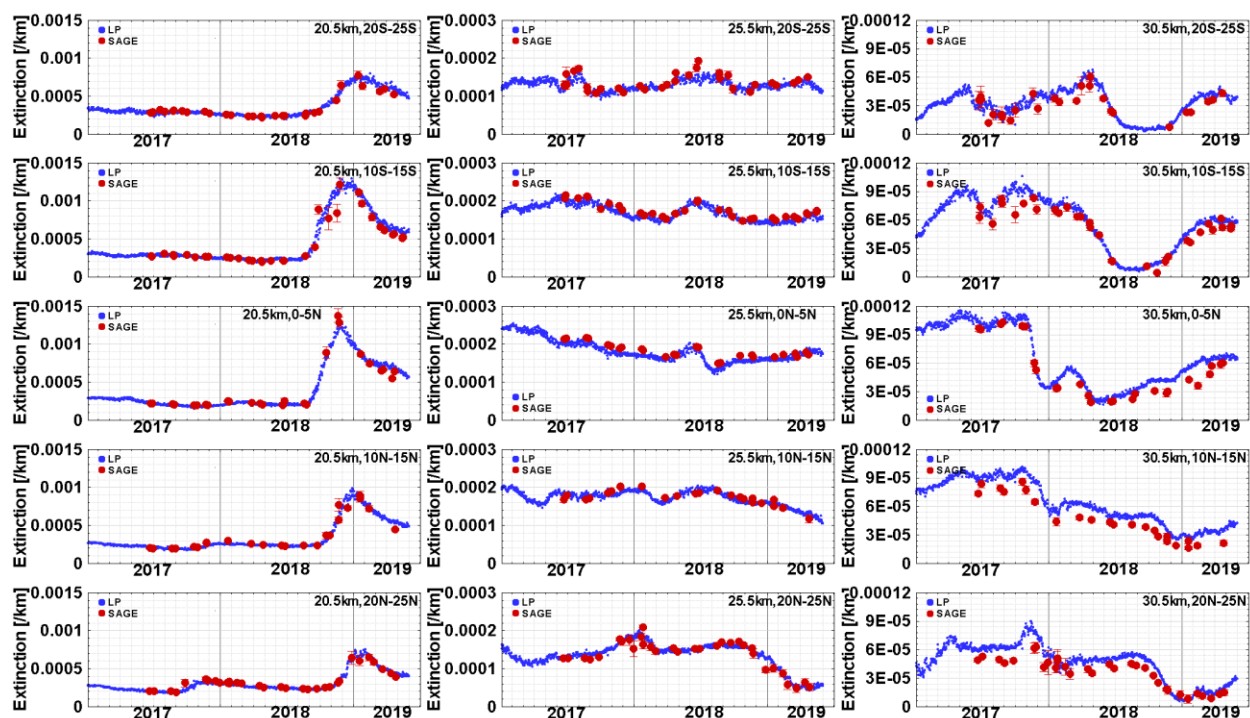

**Figure 4.** Comparison of the daily zonal mean time series of LP v1.5 (blue) and SAGE III/ISS v5.1 (red) extinction values at 675 nm for altitude at 20.5km (left column), 25.5km(middle column) and 30.5km (right column) for 5 degree latitude bands in the tropics (20°S–25°S, 10°S–15°S, 0°–5°N, 10°N–15°N, and 20°N–25°N, from the top to bottom) during the period June 2017 – May 2019. Vertical error bars on the SAGE measurements indicate standard error of the mean, σ/$\sqrt{N}$.

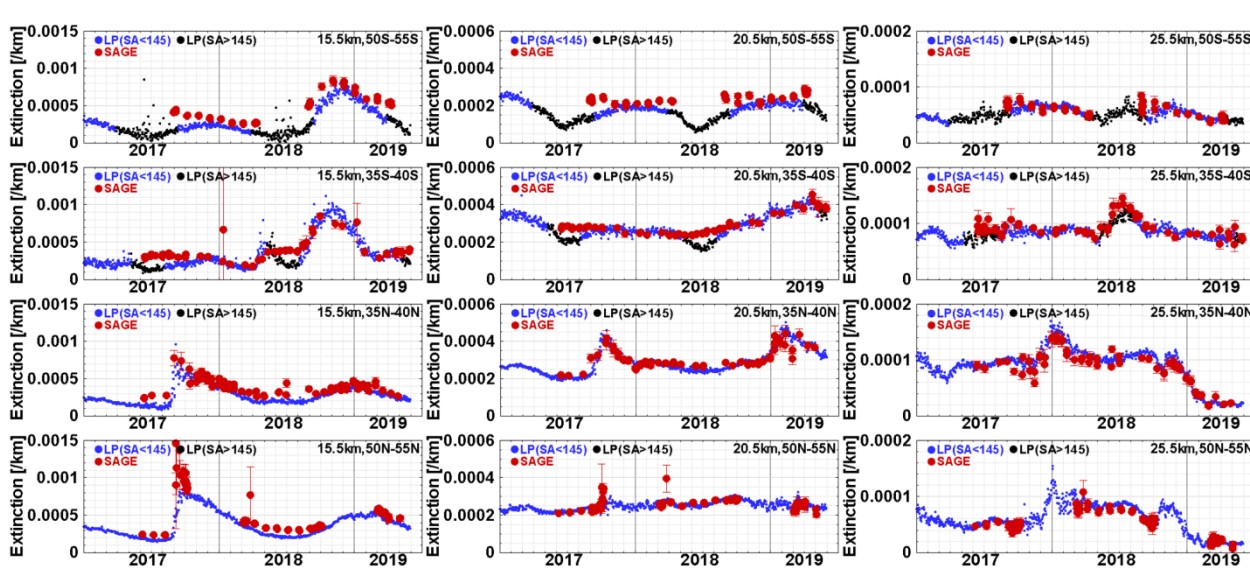

**Figure 5.** Same as Figure 4, but for altitudes at 15.5km (left), 20.5km(middle) and 25.5km (right) for latitude bands outside the tropics in 50°S–55°S, 35°S–40°S, 35°N–40°N and 50°N–55°N (from the top to bottom). The blue and black dots represent LP extinction values at scattering angles (SA) < 145° and > 145°, respectively. The LP extinction below 20 km at southern latitudes (35°S-55°S) shows the presence of erroneous seasonal variations due to lack of sensitivity to aerosol when SA > 145°.

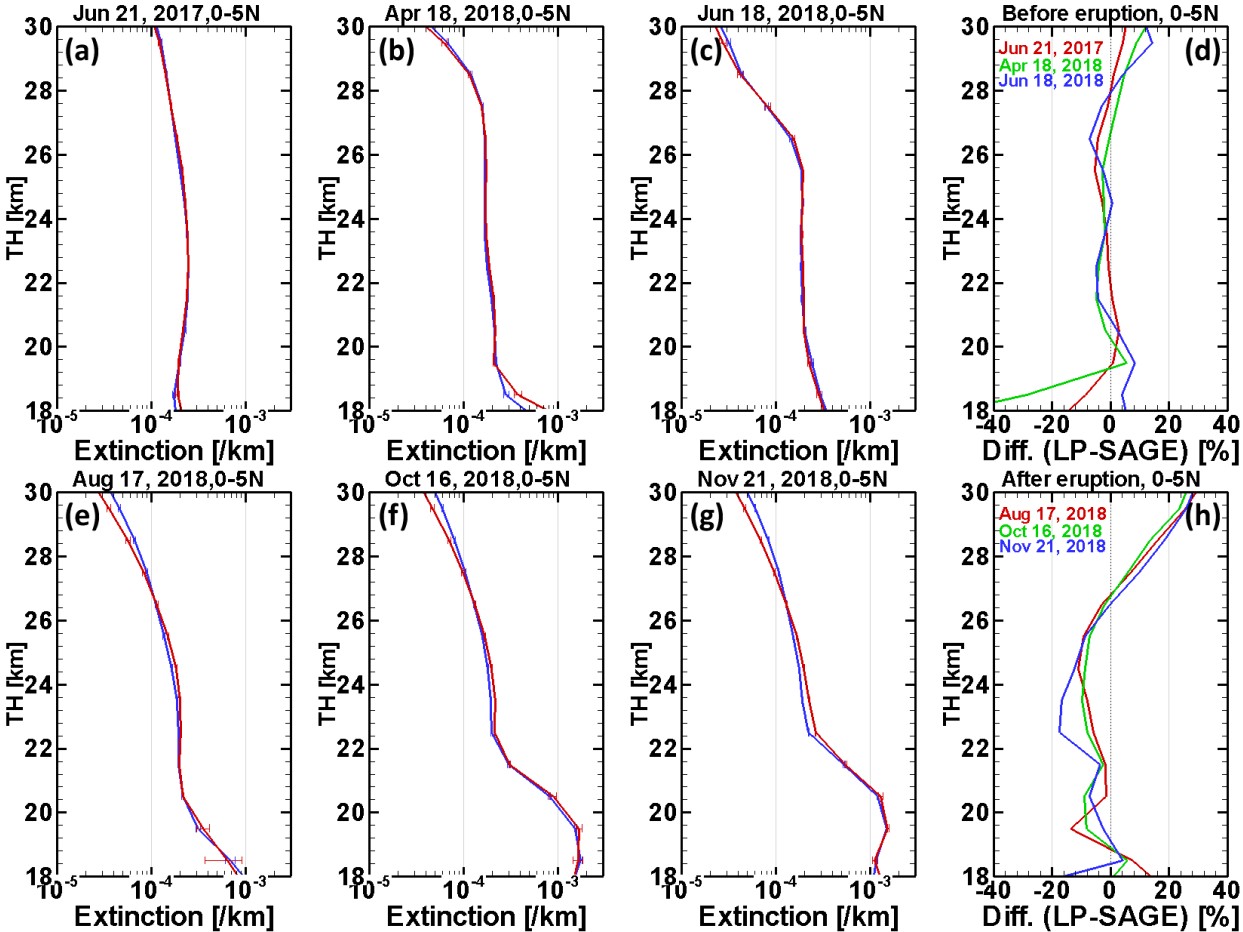

**Figure 6.** Comparison of daily zonal mean of OMPS/LP (blue) and SAGE III/ISS (red) aerosol extinction profiles in latitude bin 0-5°N for six individual days prior to (**a–c**) and after (**e–g**) the Ambae eruption on July 27, 2018. The corresponding relative differences between LP and SAGE are also shown in (**d**) and

10    (**h**). The horizontal error bars on the mean extinction profiles indicate standard error of the mean, $\sigma/\sqrt{N}$. Enhanced extinction values around 20 km after the eruption (**e–g**) are observed compared to (**a–c**).

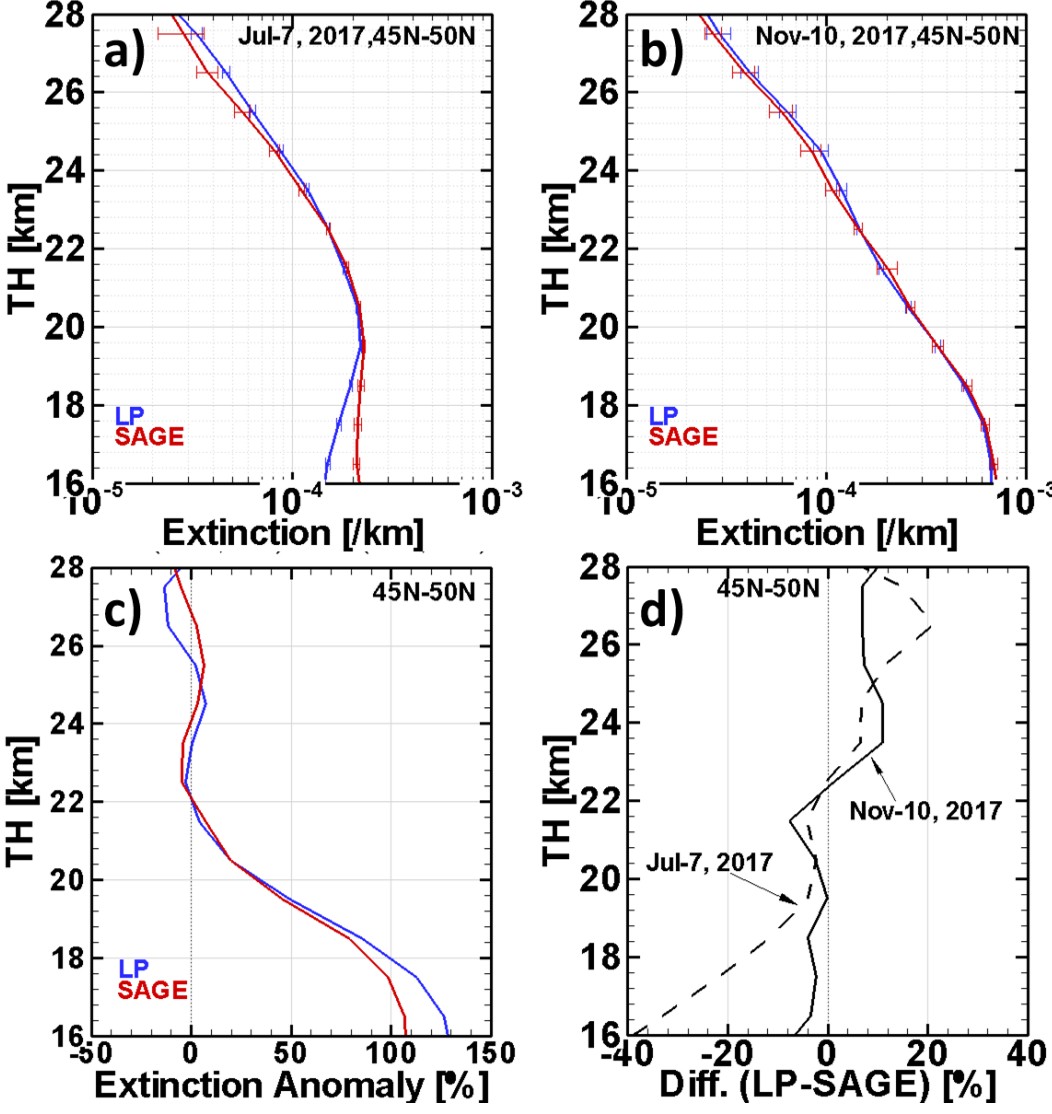

**Figure 7.** Comparison of daily zonal mean of OMPS/LP (blue) and SAGE III/ISS (red) aerosol extinction profiles in the latitude bin 45° N – 50° N: (**a**) before (July 7, 2017), (**b**) after (November, 10, 2017) the Canadian PyroCb event in late 2017, (**c**) extinction anomaly (defined as deviation from the before case)
10      and (**d**) the corresponding relative differences between LP and SAGE. The horizontal error bars on the mean extinction profiles indicate standard error of the mean, σ/$\sqrt{N}$. Enhanced extinction values below 22 km after the fire are observed in (**c**).

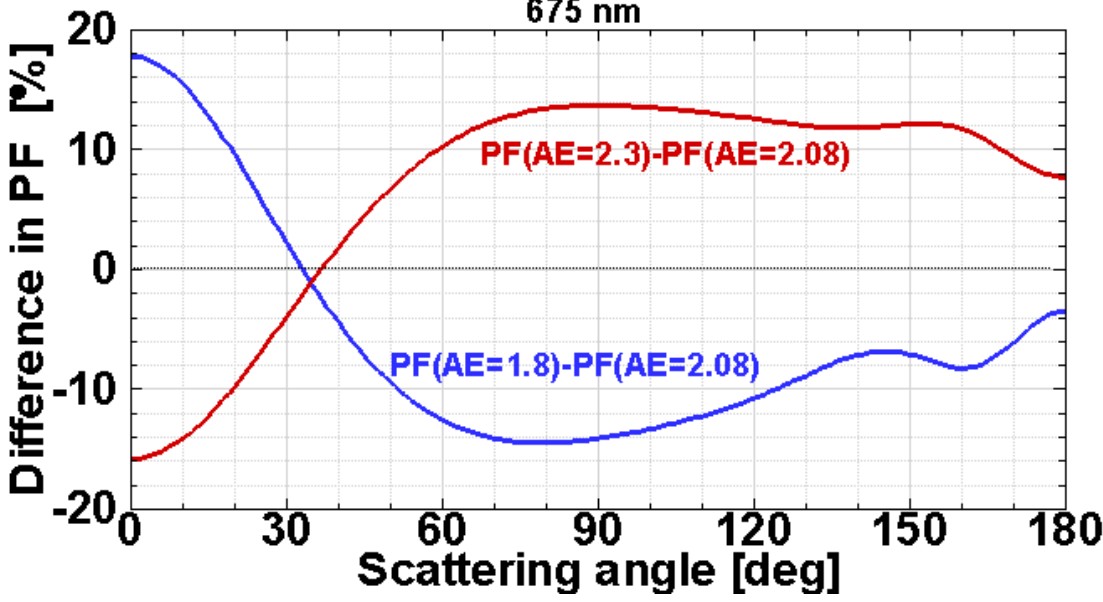

**Figure 8.** This figure shows how the aerosol phase function (PF) at 675 nm changes when the AE is
10   perturbed by ±10% of the baseline value of 2.08.

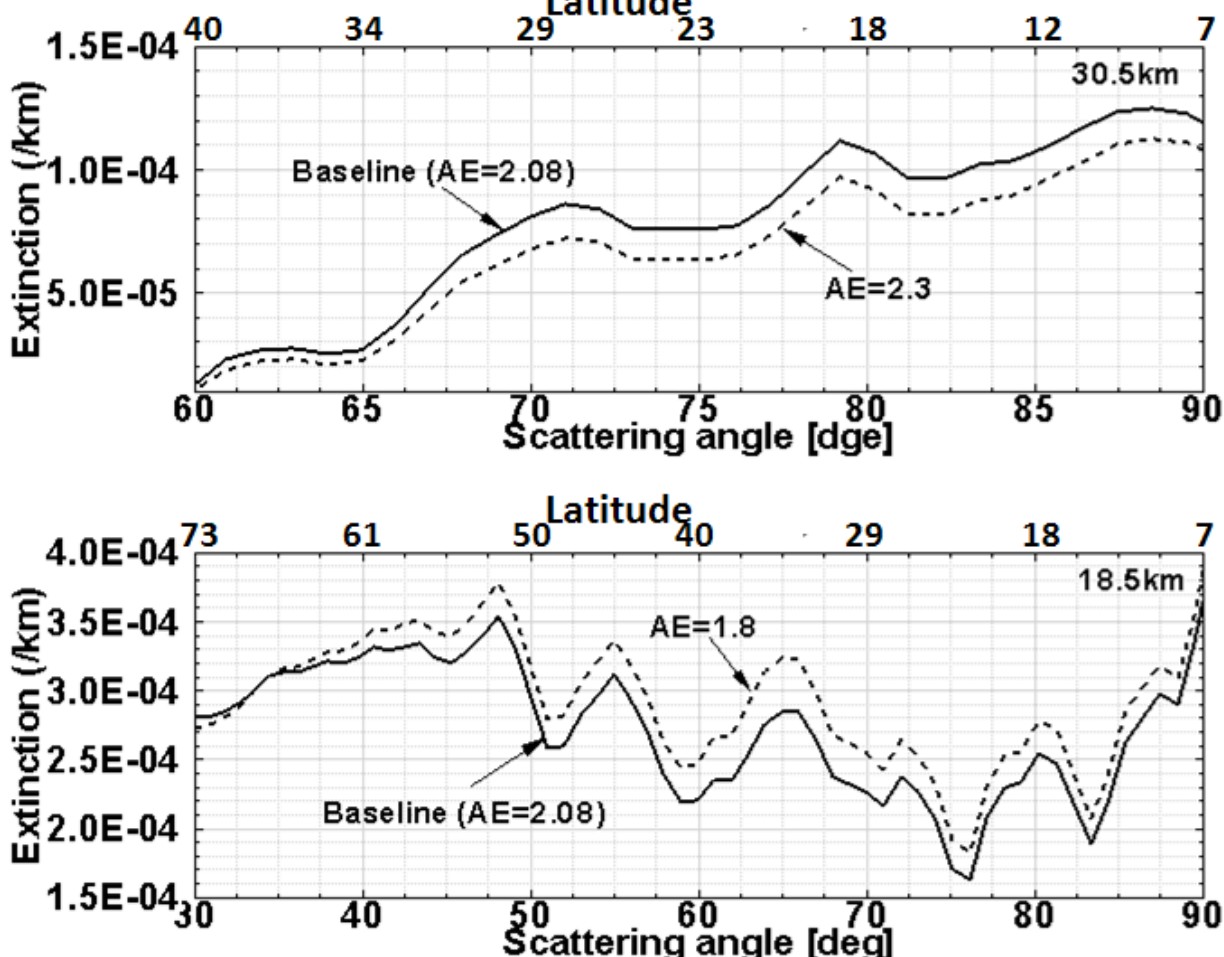

**Figure 9.** LP retrieved extinction values from the baseline particle size distribution (solid line) and from
10     the same with Angstrom Exponent (AE) adjusted by ±10% (dashed line) for altitude at 30.5 km (top
panel) and 18.5 km (bottom panel) as a function of scattering angle (x-axis bottom) and latitude (x-axis
top) using OMPS/LP measurements for a single orbit on September 12, 2016. While extinction values at
30.5 km with AE=2.3 are lower than the baseline, extinction values at 18.5 km with AE=1.8 are higher
than the baseline.

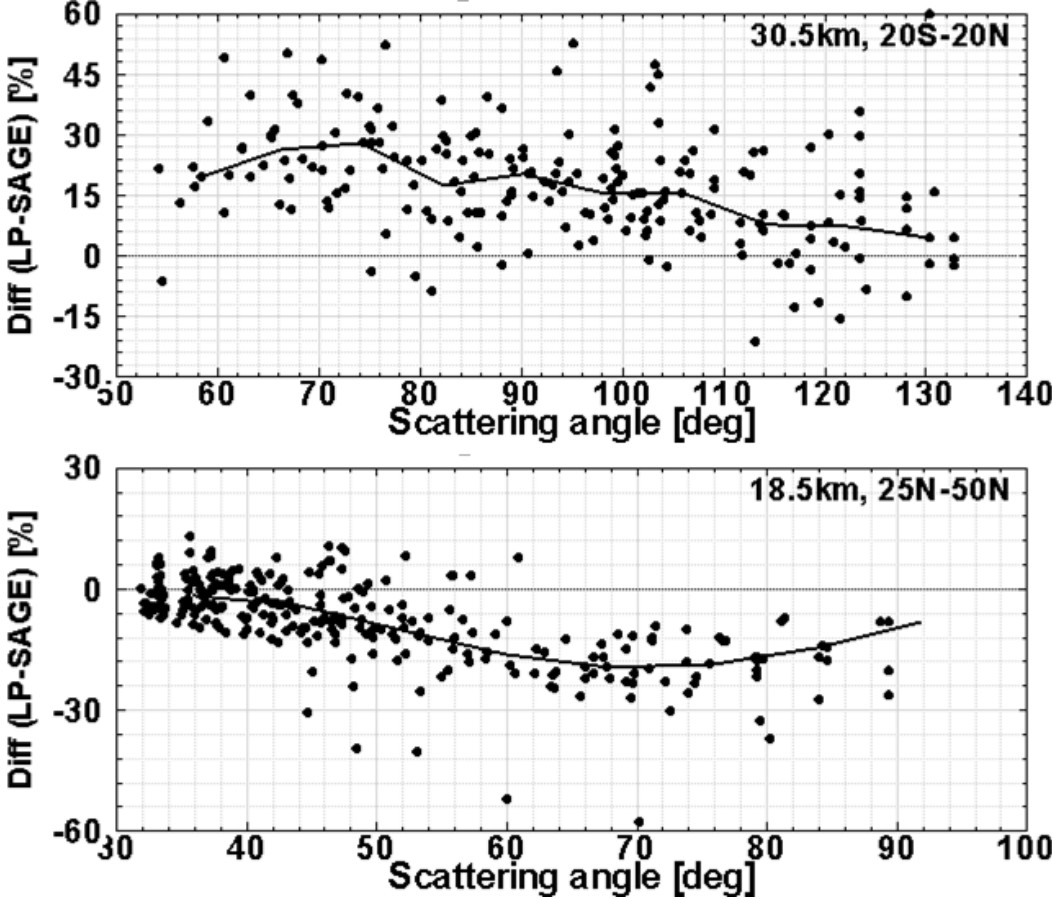

10 **Figure 10.** Relative differences in aerosol extinction between LP and SAGE III/ISS along with the median difference (solid line) as a function of LP scattering angle for altitude at 30.5 km in latitude bin 20° S - 20° N (top) and 18.5 km in latitude bin 25° N - 50° N (bottom) during the period of June 2017 - May 2019. The differences in aerosol extinction values vary significantly with scattering angle between 50–100 degree.

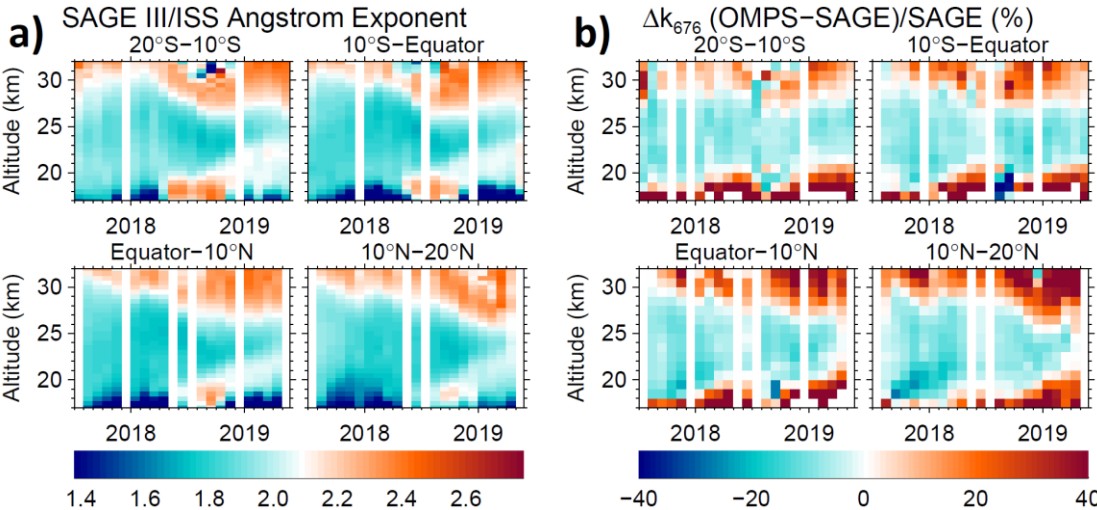

**Figure 11.** (**a**) Monthly mean AE and (**b**) differences between SAGE III/ISS and OMPS/LP extinction values at 675 nm in % as a function of altitude and time for four latitude bands in the tropics. The AE is derived from vertically smoothed SAGE III/ISS data using the aerosol extinction values at 520nm (interpolated) and 1020nm (reported) plotted on a color scale with the baseline value of 2.08 used for the OMPS/LP algorithm at its center.

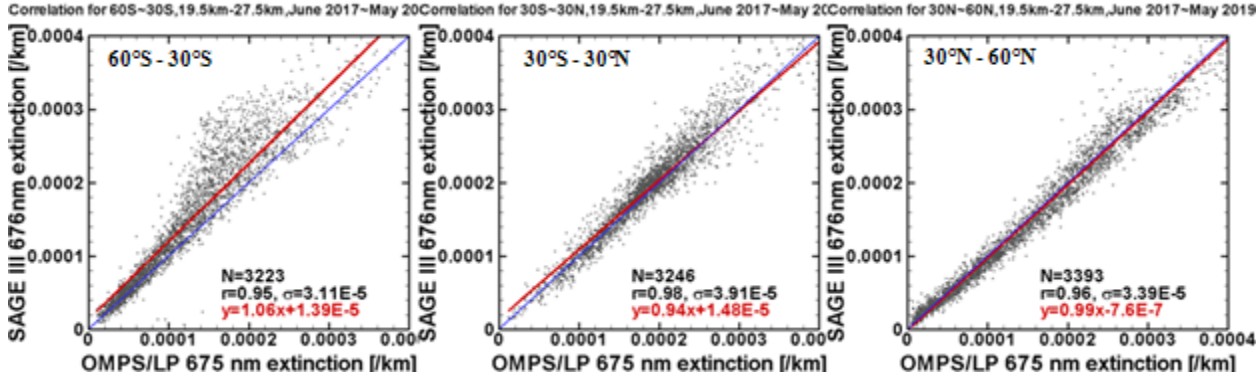

10     **Figure 12.** Scatter plot of SAGE III/ISS and OMPS/LP daily zonal means of extinction values between
19.5–27.5 km for 60°S - 30°S (left), 30°S - 30°N (middle), and 30°N - 60°N (right) for the entire time
period of June 2017~ May 2019 illustrating the correlation between the two instruments. The red line
shows the linear regression between the data points, and the blue line represents a 1:1 relationship. The
correlation coefficient r, the standard deviation of the differences (x-y) σ, and the number of elements N
15     used to compute r are also shown.

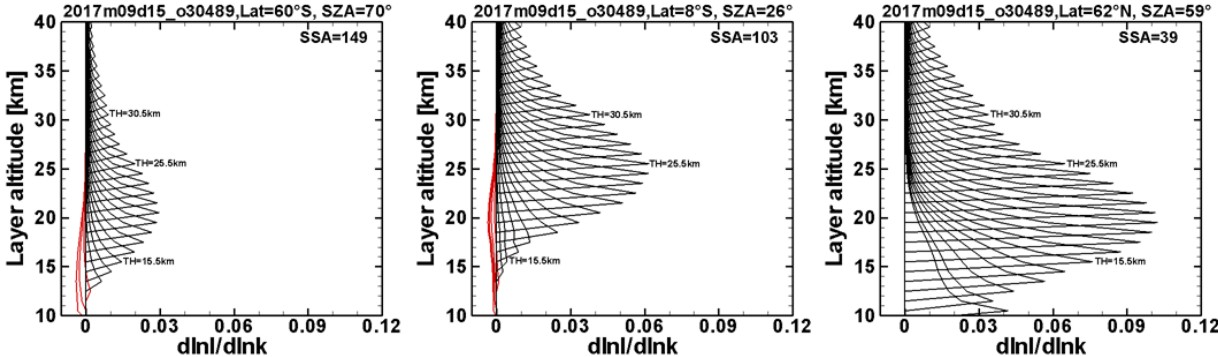

10  **Figure 13.** Example of aerosol weighting functions at wavelength 675 nm for altitude at 60° S (left), 8° S (middle) and 62° N (right). Each curve represents the sensitivity of the modelled LP radiance, *I,* at a given tangent height (TH) altitude, to a typical aerosol extinction *k* at each 1 km altitude layer from 10 to 40km. The weighting functions are positive (black line) and peaked at different altitude ranges. At lower TH altitudes and for larger SA, the 675nm weighting functions become negative (red line) and lose sensitivity
15  to aerosol extinction at lower TH altitudes.

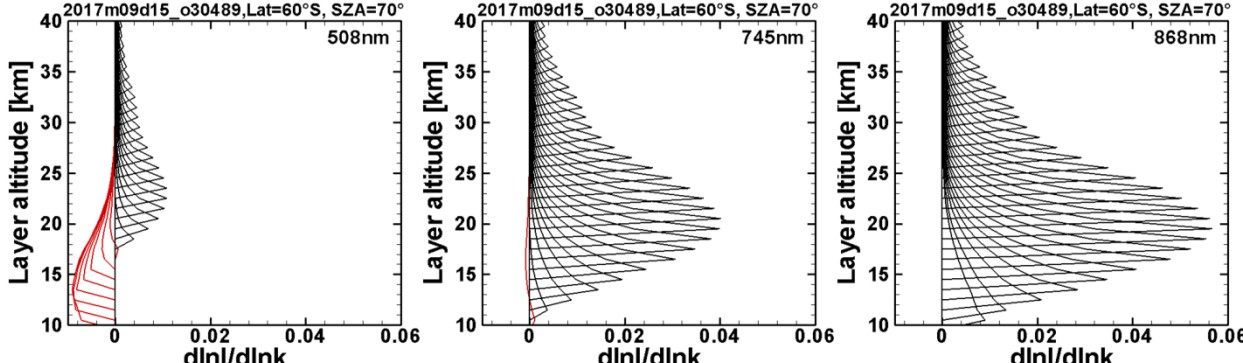

**Figure 14.** Example of aerosol weighting functions at 60° S for wavelengths at 508 nm (left), 745 nm (middle) and 868 nm (right). Each curve represents the sensitivity of the modelled LP radiance, *I,* at a given tangent altitude, TH, to a typical aerosol extinction *k* at each 1 km altitude layer from 10 to 40 km. The weighting functions are positive (black line) and peaked at different altitude ranges. At lower altitudes, the weighting functions at 508 nm and 745 nm become negative (red line) and lose sensitivity to aerosol extinction.

