# Peer review of "Evaluation of OMPS/LP Stratospheric Aerosol Extinction Product Using SAGE III/ISS Observations"

_Atmospheric Measurement Techniques, 2019_

## Referee Comment (RC1) · Anonymous Referee #1 · 4 Dec 2019

General Comments

The authors propose a similar study as in a previous paper (Chen et al., 2018) based on a much larger temporal range and enriched datasets. We could thus expect a more in-depth analysis of the behaviour found in the LP dataset. The result as appears in Section 4 is particularly disappointing in this respect: the analysis is superficial, poorly grounded, and sometimes reduced to truisms. Some statements are potentially misleading and might propagate wrong information if they are further cited without too much care in future publications. An appropriate uncertainty assessment should play a central role in such an evaluation, but there is in this paper no estimate or use of the uncertainties on the measurements considered in the intercomparisons, except for the standard deviation of the binning. The term "uncertainty" is used several times, but as

a vague concept without serious analysis.

This should be clearly improved before a possible publication.

Specific comments

L. 26-30, p.1: Aren't these two sentences telling the same thing ?

L. 26-28, p.1: Do the authors mean: "In this altitude range, the slope parameter (. . .) ? This precision might be useful in view of what is say from l. 31. L. 31-32, p.1: It would be useful to quantify the biases using the results mentioned in Section 4.3.

L. 2-3, p.2: And what about the influence of the viewing configuration ?

L. 21-25, p.3: Following Figure 1, the AE of the fitted ASD (=2.08) roughly correspond to the values found by SAGE II between 20 and 25 km in the post-2000 period, which is characterized by a load volcanic background. It should be noted that in almost all other cases (including the 1989-1990 period, also characterized by a very low aerosol content), SAGE II values are significantly lower. Hence, we can expect that this choice biases the results obtained in the case of medium to high volcanic load. This should be mentioned.

L. 27-30, p.3: This variability also and more significantly depends on the aerosol load, in the stratosphere, especially if a large time period is considered as on Figure 1 !

L. 2-3, p.4: This sentence requires a reference.

L. 5-6, p.4: It would be useful to specify why the authors foresee such a limitation on the SZA.

L. 3-4, p.5: What do the authors mean by "using SAGE samples that correspond to the OMPS/LP 1 km altitude grid" ?

L. 5-7, p.5: In this work as in Chen et al. (2018), the fact that the datasets are zonally averaged over the whole longitude range leads to neglecting the signature of local

events such as moderate volcanic eruptions. Such kind of comparison is thus not sufficient to assess the ability of the OMPS retrieval to quantify correctly extinction in case of medium to large volcanic eruptions, because their signature is diluted in the whole considered event population. In particular, the authors cannot assess adequately the effect of the weakness of the limitation of the ASD to the case of low aerosol content (See comment on l. 21-25, p.3) when using such zonal averaging. This should be at least mentioned, and either the authors should clearly mention that they restrict the scope of their paper to background aerosol situations, or they should reproduce their study over averaging windows limited in longitude and latitudes, and centered on important volcanic eruptions that occurred during the measurement period.

L. 7-10, p.5: Does it mean that the dataset corresponding to the side slits is actually not validated ? The authors should specify if this is the case or not (with some reference if relevant), and discuss the impact of this limited intercomparison exercise on the validation of the OMPS dataset.

L. 13, p.5, L. 28, p.10: What does "MLR" mean ?

L. 11-15, p.5: These references are particularly poor. In particular, Thomason and Taha (2003) only mention that "the aerosol product is produced as a residual following the removal of the effect of ozone and other species", and nothing more about the methodology. Overall, the mention about biases appears much less as a strong and well-founded statement in Thomason and Taha (2003) than what seems in the present paper. The authors should provide another reference with more solid grounds to support the discussion ?

L. 17, p5: The authors should specify the type of interpolation used here. Is it just a linear interpolation ?

Caption of Figure 3: The authors should be more specific than "below 21 km". What is the lower altitude they consider ?

Figure 3 and L. 1 p.6: It is quite striking how LP and the smoothed+interpolated SAGE profile (supposed to be particularly reliable with removed influence of ozone) are similar above 19.5 km, and much more disagree below this altitude. Which influence should be seen here ? Clouds, other aerosol types (e.g. dust) ? Or again, thicker aerosol particles as expected at such altitude (See e.g. Deshler et al., J. Geophys. Res, 108,(D5), 4167, 2003; and Bingen et al., J. Geophys. Res., 109, D06201, 2004) ? The link should also be made with the discussion in Section 4.3 .

Figure 3: What about the error bars ?

Figure 4: In view of the importance of the aerosol characterization in the upper troposphere and lower stratosphere (UTLS), it would be very useful and interesting to extend such kind of comparison to lower altitude, even if the agreement might be less impressive at these altitudes, as one can suspect from Figure 3.

L. 9-12, p.6: Again, since the binned curves compared here are likely to cover a much larger extend than the one of the volcanic cloud, even if the signature of the Ambae eruption is clearly visible here, the extinction profiles are probably a not representative of the extinction in the aerosol cloud. Consequently, I don't think that anything can be concluded here about the adequation of the assumed ASD to describe medium to high volcanic aerosol cases. Hence, I find the statement that an agreement within 20% between LP and SAGE III/ISS shows that the assumed ASD is "reasonable" in this case, really premature. See also comments on L. 5-7, p.5.

L. 15-19, p.6: Why would the dependence in temperature and humidity affect more the agreement between both datasets than the use of the assumption of "background aerosols" at lower height (e.g. 20.5 km) ? Hasn't this dependence in the atmospheric conditions a lower-order impact on the LP retrieval than the choice of ASD ? The authors should also explain why they expect that this effect of "wrong assumption" on the index of refraction would affect LP more than SAGE III/ISS.

L. 13-14, p.6: This statement and the references to the papers by Pumb and Bell (1982)

and Thomason et al. (2008) is not sufficiently grounded. The time period here is hardly longer than 2 year and is totally insufficient to assess any QBO effect. The only thing Plumb and Bell are referring to in their 1982 paper about the altitudes around 30 km is, to the best of my knowledge, that this height roughly corresponds to the maximum zonal wind amplitude of the QBO. And Thomason et al. (2008) mention in only one sentence, without any further discussion, that they observe significant aerosol variability with a period similar to the QBO at 30 km. As a conclusion, nothing convincing in the cited works seems to support the statement made here. If the authors find anything in the behaviour observed at the 5 considered latitude bands likely to reflect any QBO influence, this should be discuss appropriately. Otherwise, the authors should remove this sentence. And both citations should be removed unless the authors are able to formulate some arguments showing that they are really relevant in this particular context.

L. 21-23, p.6: This sentence is particularly empty. What do the authors mean by "very similar" ? The very good agreement found at 20.5 km in Figure 4 is much less obvious in Figure 5; the disagreement found at 30.5 km might be more important in Figure 4 than in Figure 5, although a clear lack of data in the SAGE III/ISS doesn't allow any conclusion in some cases. And the situation at 25.5 km might show a kind of mix of data gaps for SAGE III/ISS and increased disagreement in Figure 5. Furthermore, observing that "aerosol extinctions are highly variable in altitude and time" is particularly uninformative: what is "highly variable" ? One could argue that, as shown on Figure 5, the curves are quite flat – which is obviously a matter of choice of Y-scale.

L. 24, p.6: The fact that LP sees seasonal variations not seen by SAGE III/ISS and that this is interpreted as a weakness of the retrieval caused by "ASD errors and limitations" is a serious issue. Do I understand well that this concerns plots in the range 35°S-55°S at 20.5 km ? The amplitude of winter minima found here is particularly important (about 25%?) and is not reflected in any error bar, what is a worrying issue.

L. 25, p.6: What do the authors mean by "limitation of 675"Âă?

L. 25-27, p.6: Could the authors verify their statement about the influence of Canadian PyroCb from 2D maps ? Otherwise, they should qualify: "was most probably responsible (…)".

L. 27-29, p.6: The reduced data coverage in the case of SAGE III/ISS doesn't allow any relevant conclusion about a maximum limit of the disagreement between both datasets in the sense that the most prominent patterns found by LP are not observed by SAGE III/ISS. The authors should thus avoid such a quantitative, possibly misleading estimate of the agreement between both datasets.

L. 30, p.6-L. 2, p.7: The sentence "For SAGE, …, for OMPS/LP" seems to be an explanation of the statement "Under low aerosol condition". However, straylight contamination and sensitivity to small aerosol particles are two independent concepts.

L. 8-9, p.7: A typical daily latitudinal coverage in a given bin (if it contains any measurement) is about 0.3° for SAGE III/ISS, which is very small compared to the bin latitude resolution of 5°. The SAGE III/ISS are regularly spread over the whole longitudinal dimension, so that few points concern the region of the eruption. Did the authors check that the sampling provided by LP and mainly SAGE III/ISS, with respect to the spread of the volcanic plume, is adequate to conduct a relevant intercomparison ? Otherwise, they should repeat their comparison by using a more adequate choice of bins (e.g. a more focussed region, possibly during more days). This is very important in order to distinguish the part of algorithm performance and the part of mismatch in the differences observed here. And these aspect should be at least discussed adequately.

L. 22-24, p.7: The authors should mention the highest altitude reached by the PyroCb, in order to provide an insight about its expected impact on the aerosol profile.

L. 30, p7-L.2, p.8: What is the expected impact of the wrong aerosol type (sulfate instead of carbonaceous aerosols) and possibly inadequate choice ASD in this case ? Is this impact expected to be stronger in the case of LP than in the case of SAGE III/ISS ? This should be discussed.

L. 17-19, p.8: I guess the 1.8 values is close to the reported averaged SAGE II values at 18 km, *in the case of reduced aerosol load*. It is anyway not representative for a high aerosol burden. This should be specified.

L. 27, p.8: "the vertical variability of stratospheric aerosol properties": the authors should be more specific.

L. 31, p.8: "(…) to see if an ASD error exists": the authors should reword this strange sentence and specify what they mean.

Title of Section 4.4: The authors announce a discussion about the correlation between the extinction and the Angstrom exponent, what seems to foretell some rigourous study with appropriate calculations of the correlation between these quantities. But the reality appears to be (apparently) that (fast) conclusions are drawn from some visual examination of the similarity between two plots. The discussion falls thus short of the expectations. The methodology looks insufficient and either it should be revised, expanded or clarified, or the authors should change the title of the section. See also comment on L. 24, p.9.

L. 10, p.9: What do the authors mean by "vertically smoothed SAGE III/ISS data" ?

L. 11, p.9: Is the SAGE extinction at 675 nm processed as previously (interpolation using 2 close spectral channels) ? This should be specified.

L. 14-17, p.9: The formulation is confusing. Do I understand well that the authors interpolate the aerosol extinction at 520 nm? If the interpolation makes use of a second-order polynomial, why do they need 4 channels ? Or are they rather fitting such polynomial ? Is this approach more accurate than just deriving directly the AE using a linear fit of the aerosol extinction on a log-log scale ?

L. 17-19, p.9: This sentence is confusing and should be more accurate. The values above 2.08 found above 25 km height are not associated to the Mt. Ambae eruption, although they constitute the majority of the category of values > 2.08.

L. 19-20, p.9: AE below the baseline (=2.08 following L. 17, p.9) are all values plotted in blue colours in Figure 11a. It is very hard to believe that all these values are easily associated to clouds. What do the authors mean ? The authors should also specify where they see the influence of PyroCb.

L. . 21-22, p.9: I guess the authors draw this conclusion from Figure 11b. It might be useful to specify it for the sake of clarity. Furthermore, this negative bias, following Figure 11b, doesn't cover the whole stratosphere and the authors should be more specific. Finally, it seems from Figure 11b that the vertical extend of the negative bias decreases with time. Is this due to ageing of the instrument ? Or do the authors have another interpretation for this general trend ?

L. 22-23, p.9: Where is this statement coming from ? The difference between LP AE and AE derived from SAGE is not shown anywhere. Do they mean SAGE III/ISS AE illustrated in Figure 11a ? Otherwise, they need to show appropriate results to support their statement.

L. 24, p.9: There is of course a strong correlation between AE and ASD, but these quantities are still different. Furthermore, visually, the altitude range takes roughly blue colours in similar regions of both plots in Figure 11 (say, the "lower stratosphere"), but the details of the time evolution of this altitude range is different in Figs. (a) and (b) and the correlation between the negative values (the various blue tones) cannot be assessed without an adequate calculation. Therefore, saying that there is "a clear correlation" is absolutely premature (this apparent correlation might be purely fortuitous), and should be removed or clearly qualified.

L. 25-27:, p.9: Where is this statement coming from ? The only results shown about southern mid-latitudes are the ones in Figure 5, and I don't see how they could lead to these results. The authors should bring the necessary developments and/or explanations to support their conclusion, or remove this sentence.

L. 10, p.10: The extinction units are missing.

L. 11, p.10: What do the authors mean by "wavelength limitations" ?

L. 11, p.10 and Figure 13: The concept of "aerosol weighting function" is never defined in the text and should be appropriately explained.

L. 12-15, p.10: Could the authors shortly explain the reason for the different sensitivity to aerosol in the northern mid-latitudes and tropical latitudes, and the southern mid-latitudes ?

L. 4-5, p.11: See comments on L. 24, p.6 and L.24, p.9. I don't agree about the conclusion on the robustness of the measured extinction variability.

L. 8-10, p.11: Measurement uncertainties are never discussed nor quantitatively mentioned before.

L. 19, p.11: "differing" is not an adequate term. What is a "differing good agreement" ?

L. 4-7, p.12: These aspects are discussed in a paper by Malinina et al. (2019), amt, 3485-3502. It seems appropriate to cite this paper.

L. 7-9, p.12: "Lower sensitivity" of what ? Which kind of uncertainties are the authors talking about ? To which specific (and different) concepts do the authors refer by "lower sensitivity" and "reduced retrieval accuracy" ? The increased extinction uncertainty above 28 km and below 19 km mentioned in L. 5, p.9 seems to refer to retrieval uncertainty, while it is not clear if the uncertainty cited in L. 15 and 18, p.10 and linked to "noise amplification" refers to instrumental noise or retrieval noise. The discussion on uncertainty assessment is clearly insufficient.

L. 10, p.12: What do the authors mean by "straylight contamination is more obvious" ?

L. 11, p12: The concept of "random discrepancy" looks strange.

L. 11-12, p.12: This sentence should be qualified. It is not clear whether the authors consider the association between the large discrepancy between LP and SAGE, and the presence of clouds as obvious (in my opinion, it is not), or as a working hypothesis.

Abstract and summary/conclusions: It seems appropriate and important to explicitly mention the presence of erroneous "seasonal variations" in the OMPS-LP dataset in the abstract and in the summary and conclusions, since this behaviour risks to induce misinterpretations in future works.

Technical comments

General remark: The authors make use of both SAGE II and SAGE III/ISS datasets. Sometimes, they refer to "SAGE". They should be more specific.

General remark: I guess the correct spelling is "mid-latitude" instead of "midlatitude".

L. 22, p.3: The units should be corrected (microns instead of m).

L. 22, p.6, L23, p.8, Caption Figure 11, l.11, p.9, etc.: "extinction" is a physical parameter and should be singular. If the authors want to use a plural form, they should use "extinction values". Please check the whole document.

L. 30-31, p.6: The expression "The AE is quite scattered" seems improper to me. AE is a physical property.

L. 30, p.6: I don't think that the formulation "smaller aerosols" is appropriate. Aerosol is a substance in suspension in the air. The authors should use the term "particle" that can be associated to the concept of size.

L. 9, p.9: missing punctuation.

Caption Figure 11: It is confusing that the labels (a) and (b) are after the corresponding part of the caption. The authors should put the labels first.

L. 29, p.10: The correct expression for "Chappius" is "Chappuis bands". The authors should use the correct one.

L. 4, p.11: Which time series ? Please be specific.

L. 6, p.11: I guess there is only one retrieval ? And "Impact" seems more appropriate

than "impacts".

L. 9, p.12: duplicated word.

---

## Referee Comment (RC2) · Anonymous Referee #2 · 16 Dec 2019

The Stratospheric Aerosol Layer is an important component of the earth atmosphere through its impacts on climate and stratospheric ozone especially after large volcanic eruptions. Chen et al. (2019) evaluate the OMPS/LP stratospheric aerosol product using the SAGE III/ISS observations. They found a good agreement (+/-25%) between OMPS/LP and a modified version of the SAGE III V5.1 data between 20 and 28 km. OMPS/LP and SAGE III/ISS data are analyzed after a moderate volcanic eruption and extreme fire reaching the stratosphere to highlight the contribution of those events on the stratospheric aerosol extinction. Finally, the sensivity of the aerosol retrieval to the assumed size distribution is also investigated at the end of the study.

I have a number of major concerns about this paper before it can be published in AMT :

[Figure]

1- Introduction The introduction does a very poor job in explaining why this work is important and why the stratospheric aerosol layer should be studied. I suggest the authors to do a literature overview of this topic to explain.

2- Novelty of this study? As mentioned by the authors, a precedent paper was published last year to evaluate the new OMPS/LP aerosol product (V1.5) with the SAGE III/ISS data. This study aims to extend this analysis with one more year observations but do not further explain the scientific justifications for providing this update. What is so different between this paper and Chen et al. (2018) ? This is not justified with the publication of a new algorithm or new version of the OMPS dataset so why is it important to publish this ?

3- Justification for using CARMA in April 2012. An important part of the retrieval is the assumption of size parameters into the radiative transfer model to infer the aerosol extinction at 675 nm from OMPS/LP. The description of the algorithm (section2) provides the basis to understand how the extinction is inferred. A gamma size distribution is used to fit size distribution from the CARMA aerosol module running with GEOS model in April 2012. I have several questions associated with this approach:

- Why do you use a Gama function to fit the model data? Bi-Lognormal distributions have been commonly used to fit stratospheric aerosol data such as those observed by the University of Wyoming for more than 30 years (Deshler et al., 2003) - It's rather strange to use one month of model data as an input to constrain a retrieval algorithm. Moreover, the caveat is that the satellite output data will not able to be used by modelers using CARMA-GEOS since they are not independent to each other.

4- Modification/improvement of the SAGE III/ISS official product. In order to correct an apparent issue with the 675 nm extinction coefficient from SAGE III/ISS due to interference with ozone, the authors developed a new algorithm to interpolate the 675 nm channel data using 449 and 756 nm. Without providing further validation of this technique, the authors acknowledge that the new retrieved 675 nm coefficient from

SAGE III is used for comparison and to some extent validate OMPS-LP. I think the approach is questionable here: Without further validation of the new retrieved 675nm extinction coefficient from SAGE III/ISS, you assume that it will be your new reference to compare and validate OMPS-LP. I think OMPS-LP should first be validated/compared with the official product from SAGE III/ISS at 675 nm before transforming the SAGE III data.

Comments:

1) P1-l16: "has been flying". I believe that this expression could be improved in a scientific publication 2) P1-l29: "high degree of correlation". Quantify here. 3) P1-L32: "systematically lower…" You are not measuring the same air masses so the different between the two instruments are expected to be high at the tropopause or below. 4) P1-L34: "altitude dependent..." Not only altitude but also latitude. 5) P2-L3: "cloud contamination": That is of the main issue, which is poorly discussed in this paper. 6) P2-L8: " (Ridley et al., 2014) Âń. There is a very poor review of the available literature on this topic. This should be improved. 7) P2-26: "has become operational..." What does it mean here ? 8) P2-L27: "..A more comprehensive..": Does it justify another publication on OMPS-LP? 9) P4-L10: "The SAGE III/ISS developed..": Something is missing between SAGE III/ISS and the verb. It does not read well. 10) P5-L10: "Cloud height rejected…": How is cloud top height inferred from OMPS-LP? 11) P6-L19: "Figure 11..": You should remove a reference to a figure that you do not explain at this stage of the paper. 12) P7/Figure7. Figure 7 does not really highlight nicely the emergence of new stratospheric aerosol layers before and after the Ambae eruption and the fire in Canada. I would rather suggest producing an anomaly plot before and after each event. 13) P7-10. How sure are you that the corresponding increase in extinction is associated with this eruption? Provide reference or further analysis to make your point. 14) P7-L23. "were produced.." use a better verb than "produce" here. 15) P8-L8. "..for the main aerosol layer.." What do you mean by "main layer", the Junge layer ? The stratospheric aerosol reservoir in the tropics? Be more accurate so that the reader

can understand what you're talking about. 16) P8-L27-28: "The results. . ." I do not understand this sentence. Please rephrase and improve. 17) P9-L18: "are easily associated.." It does not read well in English. Please improve. 18) P9-L19: You need to include references here. 19) P12-L12: "..broken clouds.." What do you mean by broken clouds, cirrus clouds ?

———————————————

---

## Author Comment (AC1) · 30 Jan 2020

Reply to Reviewer 1 Z. Chen et al. zhong.chen@ssaihq.com

We thank Reviewer #1 for reading our paper in detail and providing useful comments and suggestions. Below we answer the reviewer's concerns and make the necessary corrections to the paper.

General Comments: The authors propose a similar study as in a previous paper (Chen et al., 2018) based on a much larger temporal range and enriched datasets. We could thus expect a more in-depth analysis of the behaviour found in the LP dataset. The result as appears in Section 4 is particularly disappointing in this respect: the analysis is superficial, poorly grounded, and sometimes reduced to truisms. Some statements are

potentially misleading and might propagate wrong information if they are further cited without too much care in future publications. An appropriate uncertainty assessment should play a central role in such an evaluation, but there is in this paper no estimate or use of the uncertainties on the measurements considered in the intercomparisons, except for the standard deviation of the binning. The term "uncertainty" is used several times, but as a vague concept without serious analysis. This should be clearly improved before a possible publication.

Reply: We thank Reviewer #1 for a careful review of this paper. We believe that the present study is a more in-depth analysis of the behavior found in the LP dataset. The central scope of the paper is the evaluation of the LP algorithm performance for background aerosol situations. Our analysis of SAGE III/ISS data specifically addresses possible biases with OMPS/LP results arising from differences in vertical resolution and possible ozone contamination. The measurement uncertainties which discussed in details in our previous papers (Loughman et al., 2018; Kramarova et al., 2018) are estimated to be < 1%. Since the small measurement uncertainties have weak impact on the intercomparisons, we focus on the retrieval uncertainties which play a central role in the evaluation of the algorithm performance. We believe we have appropriately assessed the retrieval uncertainties in Section 4. We now add text to mention the measurement uncertainties (see Reply to Comment on L. 30, p.6-L. 2, p.7). In any cases, we should have made some statements much more clear in our manuscript and have now done so. We answer the points one-by-one below.

Specific comments L. 26-30, p.1: Aren't these two sentences telling the same thing?

Reply: The first sentence highlights the results of statistical analysis. The seconded sentence told us that LP can capture the variability of stratospheric aerosol layer well. We rewritten the second sentence as "Comparisons of extinction time series show that the both instruments capture the variability of stratospheric aerosol layer well. The differences between the two instruments vary from 0% to ±25% depending on altitude, latitude and time."

L. 26-28, p.1: Do the authors mean: "In this altitude range, the slope parameter (. . .)? This precision might be useful in view of what is say from l. 31.

Reply: Yes, we have added "In this altitude range" to the beginning of the sentence.

L. 31-32, p.1: It would be useful to quantify the biases using the results mentioned in Section 4.3.

Reply: We have added "with significant biases up to $\pm13\%$".

L. 2-3, p.2: And what about the influence of the viewing configuration?

Reply: We have added "the influence of the viewing configuration".

L. 21-25, p.3: Following Figure 1, the AE of the fitted ASD (=2.08) roughly correspond to the values found by SAGE II between 20 and 25 km in the post-2000 period, which is characterized by a load volcanic background. It should be noted that in almost all other cases (including the 1989-1990 period, also characterized by a very low aerosol content), SAGE II values are significantly lower. Hence, we can expect that this choice biases the results obtained in the case of medium to high volcanic load. This should be mentioned.

Reply: We have modified the sentence: "... during the period 2000-2005, which is characterized by a low volcanic background ... Other time periods known to have low aerosol loading (e.g. 1989-1990) show lower values of AE in the SAGE II dataset. Thus, the reference ASD adopted here for LP retrievals may produce a bias in extinction values derived during medium or high volcanic aerosol loading."

L. 27-30, p.3: This variability also and more significantly depends on the aerosol load, in the stratosphere, especially if a large time period is considered as on Figure 1!

Reply: We have added text "This variability also and more significantly depends on the aerosol load, in the stratosphere, especially if a large time period is considered as in Figure 1"

L. 2-3, p.4: This sentence requires a reference.

Reply: We have included a reference: "(Chahine, 1970)".

L. 5-6, p.4: It would be useful to specify why the authors foresee such a limitation on the SZA.

Reply: OMPS LP only measures scattered sunlight (which limits observations to SZA < 90 degrees), and the radiative transfer algorithm used for LP retrievals becomes less accurate at very high SZA, so we cut off retrievals at SZA = 88 degrees to limit that possible source of error. We modified sentence: "For the LP aerosol product, retrievals are only performed for daytime observations (solar zenith angle SZA < 88°)".

L. 3-4, p.5: What do the authors mean by "using SAGE samples that correspond to the OMPS/LP 1 km altitude grid"?

Reply: This sentence has been removed as it is not essential.

L. 5-7, p.5: In this work as in Chen et al. (2018), the fact that the datasets are zonally averaged over the whole longitude range leads to neglecting the signature of local events such as moderate volcanic eruptions. Such kind of comparison is thus not sufficient to assess the ability of the OMPS retrieval to quantify correctly extinction in case of medium to large volcanic eruptions, because their signature is diluted in the whole considered event population. In particular, the authors cannot assess adequately the effect of the weakness of the limitation of the ASD to the case of low aerosol content (See comment on l. 21-25, p.3) when using such zonal averaging. This should be at least mentioned, and either the authors should clearly mention that they restrict the scope of their paper to background aerosol situations, or they should reproduce their study over averaging windows limited in longitude and latitudes, and centered on important volcanic eruptions that occurred during the measurement period.

Reply: This was one of the major criticisms raised by the reviewer in the overall remarks through the review. We agree that the use of zonal averaging over the whole longitude

range leads to neglecting the signature of local events. As indicated in Chen et al. (2018) and this manuscript, the LP retrieval is performed using background layer size distribution parameters, and assumed ASD is appropriate for OMPS/LP aerosol extinction retrievals in most of the stratosphere for background aerosol. We believe that our comparison is sufficient to assess the ability of the OMPS retrieval under background aerosol conditions when using such latitude zonal mean.

We clarify this by changing this sentence to read "The daily averaged data in the same latitude bin and on the same day were used for the comparison with an assumption that the variation with longitudinal bands is much smaller than the variation with latitudinal bands. For background aerosol situations, the longitudinal variation of stratospheric aerosol could be small because of efficient mixing in the zonal direction and strong horizontal transport prevailing in the region (Sunilkumar et al., 2011). In cases of medium to large volcanic eruptions, however, longitudinal differences could be large and a restriction should be placed on longitude to capture the signature of potential longitudinal variation."

We added text to the Abstract: "…were available to evaluate our ability to characterize background aerosol conditions"

We also added two sentences to the Introduction to emphasize this. "The assumed ASD is designed to represent the long-term background stratospheric aerosol loading" and "the central scope of the paper is the evaluation of the LP algorithm performance for background aerosol situations".

L. 7-10, p.5: Does it mean that the dataset corresponding to the side slits is actually not validated? The authors should specify if this is the case or not (with some reference if relevant), and discuss the impact of this limited intercomparison exercise on the validation of the OMPS dataset.

Reply: We analyzed the aerosol retrievals from left and right slits. Our internal analysis shows differences between the three slits. Those differences are relatively

small compared to the overall magnitude of the adjustments (both static and intra-orbital+seasonal) for any single slit. Only center slit results are shown for clarity, and that these results should be representative of results derived using left slit or right slit measurements. As mentioned in the manuscript, the reason why we focus on analyses of LP data from the center slit for the comparison is that the center slit has better straylight and tangent height corrections compared to the left and right slits (Moy et al., 2017; Kramarova et al., 2018). "

L. 13, p.5, L. 28, p.10: What does "MLR" mean?

Reply: This is just the short name of the product as given by the SAGE team, not an abbreviation. This is why it's in quotations. (Technically, MLR does stand for multiple linear regression because the algorithm performs it, but that's misleading because essentially all of the products derive from some form of multiple linear regression.)

L. 11-15, p.5: These references are particularly poor. In particular, Thomason and Taha (2003) only mention that "the aerosol product is produced as a residual following the removal of the effect of ozone and other species", and nothing more about the methodology. Overall, the mention about biases appears much less as a strong and well-founded statement in Thomason and Taha (2003) than what seems in the present paper. The authors should provide another reference with more solid grounds to support the discussion? Reply: The 2003 reference is removed as suggested. This whole concept of biases came up from internal investigations into the data, none of which is published yet. As such, the "private communication" is appropriate. The SAGE team started an ozone validation paper (currently under review in JGR:A) where they brought this up in the context of how apparent biases in the aerosol spectrum likely originate from how ozone is retrieved and the impact on the ozone product (though the cause is ozone and the effect is aerosol). This paper does discuss the retrieval algorithm a little more. We now provide this reference: Wang, H.J., R. Damadeo, D. Flittner, N. Kramarova, G. Taha, S. Davis, A.M. Thompson, S. Strahan, Y. Wang, L. Froidevaux, D. Degenstein, A. Bourassa, W. Steinbrecht, K. Walker, R. Querel, T. Leblanc, S.

Godin-Beekmann, D. Hurst, and E.J. Hall, E.: Validation of SAGE III/ISS Solar Ozone Data with Correlative Satellite and Ground Based Measurements, JGR Atmos., 2020, in review.

L. 17, p5: The authors should specify the type of interpolation used here. Is it just a linear interpolation?

Reply: We have added "using a log-linear interpolation".

Caption of Figure 3: The authors should be more specific than "below 21 km". What is the lower altitude they consider?

Reply: We replaced "above 27 km below 21 km" by "above 19 km" in consisting with text.

Figure 3 and L. 1 p.6: It is quite striking how LP and the smoothed+interpolated SAGE profile (supposed to be particularly reliable with removed influence of ozone) are similar above 19.5 km, and much more disagree below this altitude. Which influence should be seen here? Clouds, other aerosol types (e.g. dust)? Or again, thicker aerosol particles as expected at such altitude (See e.g. Deshler et al., J. Geophys. Res, 108,(D5), 4167, 2003; and Bingen et al., J. Geophys. Res., 109, D06201, 2004)? The link should also be made with the discussion in Section 4.3. Reply: We admit we are unclear what the viewer is referring to by "thicker aerosol particles", since no that term is presented in the two references suggested by the reviewer. Perhaps the reviewer is referring to "thick aerosol layers" or "larger particle size"?

We now use the term "thick aerosol layers" and have added the following text: "The figure shows how LP and the smoothed+interpolated SAGE III profile are similar above 19.5 km, and have more disagreement below this altitude. In the upper troposphere and lower stratosphere (UTLS), particularly in the tropics, the presence of thicker aerosol layers may have an impact on the retrieved extinction levels (Kremser et al. 2016)."

This event is also mentioned in the end of Section 4.3: "Another important aspect that influences the comparison below 19 km is the presence of clouds and thicker aerosol layers discussed earlier (Figure 3)."

Figure 3: What about the error bars?

Reply: We have now added the error bars in the two panels in the figure with associated text.

Figure 4: In view of the importance of the aerosol characterization in the upper troposphere and lower stratosphere (UTLS), it would be very useful and interesting to extend such kind of comparison to lower altitude, even if the agreement might be less impressive at these altitudes, as one can suspect from Figure 3.

Reply: We expanded the comparison to a lower altitude at 15.5km in Figure 5 with relevant text as suggested. The updated figure depicts time series comparisons at15.5 km for different latitude bands outside the tropics where clouds are not an issue (see Reply to Comment on L. 5-7, p.5 in details)

L. 9-12, p.6: Again, since the binned curves compared here are likely to cover a much larger extend than the one of the volcanic cloud, even if the signature of the Ambae eruption is clearly visible here, the extinction profiles are probably a not representative of the extinction in the aerosol cloud. Consequently, I don't think that anything can be concluded here about the adequation of the assumed ASD to describe medium to high volcanic aerosol cases. Hence, I find the statement that an agreement within 20% between LP and SAGE III/ISS shows that the assumed ASD is "reasonable" in this case, really premature. See also comments on L. 5-7, p.5.

Reply: This was the same criticism in Comment on L. 5-7, p.5. We have repeated that we did not mention "medium to high volcanic aerosol cases" in this section, but we have modified the text as recommended to read the following:

"For the background aerosol condition based on the June, 2017 to June 2018 period,

LP and SAGE are within 20% of each other at 20.5 km and 25.5 km, indicating that the assumed ASD in LP v1.5 algorithm is adequate in this altitude range."

We also changed the section title to "Aerosol Extinction Variability".

L. 15-19, p.6: Why would the dependence in temperature and humidity affect more the agreement between both datasets than the use of the assumption of "background aerosols" at lower height (e.g. 20.5 km)? Hasn't this dependence in the atmospheric conditions a lower-order impact on the LP retrieval than the choice of ASD? The authors should also explain why they expect that this effect of "wrong assumption" on the index of refraction would affect LP more than SAGE III/ISS.

Reply: While the SAGE algorithm does not make any assumptions about aerosol microphysics, the LP algorithm assumes an aerosol refractive index which varies significantly with temperature (Russell et al, 1996) and, therefore, has a direct effect on the aerosol scattering phase function.

We have now added text: "While the SAGE algorithm does not make any assumptions about aerosol microphysics, aerosol extinction profiles from OMPS/LP suffer from uncertainties due to assumed ASD and refractive index"

L. 13-14, p.6: This statement and the references to the papers by Pumb and Bell (1982) and Thomason et al. (2008) is not sufficiently grounded. The time period here is hardly longer than 2 year and is totally insufficient to assess any QBO effect. The only thing Plumb and Bell are 'referring to in their 1982 paper about the altitudes around 30 km is, to the best of my knowledge, that this height roughly corresponds to the maximum zonal wind amplitude of the QBO. And Thomason et al. (2008) mention in only one sentence, without any further discussion, that they observe significant aerosol variability with a period similar to the QBO at 30 km. As a conclusion, nothing convincing in the cited works seems to support the statement made here. If the authors find anything in the behaviour observed at the 5 considered latitude bands likely to reflect any QBO influence, this should be discuss appropriately. Otherwise, the authors should remove this sentence. And both citations should be removed unless the authors are able to formulate some arguments showing that they are really relevant in this particular context.

Reply: We have removed the two citations as requested and revised the sentence as: "The large temporal variability present in both datasets at 30.5 km is suggestive of a quasi-biennial oscillation (QBO) signal, although there is a shift in phase between latitude bands."

L. 21-23, p.6: This sentence is particularly empty. What do the authors mean by "very similar"? The very good agreement found at 20.5 km in Figure 4 is much less obvious in Figure 5; the disagreement found at 30.5 km might be more important in Figure 4 than in Figure 5, although a clear lack of data in the SAGE III/ISS doesn't allow any conclusion in some cases. And the situation at 25.5 km might show a kind of mix of data gaps for SAGE III/ISS and increased disagreement in Figure 5. Furthermore, observing that "aerosol extinctions are highly variable in altitude and time" is particularly uninformative: what is "highly variable"? One could argue that, as shown on Figure 5, the curves are quite flat – which is obviously a matter of choice of Y-scale.

Reply: We removed this sentence as it is uninformative. We also removed the comparison at 30 km from Figure 5 as it is less important than in Figure 4.

L. 24, p.6: The fact that LP sees seasonal variations not seen by SAGE III/ISS and that this is interpreted as a weakness of the retrieval caused by "ASD errors and limitations" is a serious issue. Do I understand well that this concerns plots in the range 35◦S-55◦S at 20.5 km? The amplitude of winter minima found here is particularly important (about 25%?) and is not reflected in any error bar, what is a worrying issue. Reply: We thank the reviewer for pointing out this. To address this issue, we replot data in Figure 5 by applying scattering angle filter and add a new section (4.6 limitations of Wavelength). The paragraph in Section 4.1 has been rewritten to be "Figure 5 shows the time series comparison between OMPS/LP and SAGE III/ISS measurements at

15.5 km (left panel), 20.5 km (middle panel) and 25.5 km (right panel) for different latitude bands outside the tropics where clouds are not an issue. The blue and black dots in Figure 5 represent the LP extinction calculated at scattering angles (SA) < 145° and > 145°, respectively. Again, the highly variable nature of the stratospheric aerosols with time and latitude is well represented by the two instruments. Canadian pyro-cumulonimbus (PyroCb) was most probably responsible for increasing aerosol extinction values at 15.5 and 20.5 km in the 35°N-55°N latitude bands in late 2017, and the effect of Mt. Ambae can be seen in the 35°S –55 °S time series in late 2018. Good agreement is found between both instruments at 20.5 and 25.5 km, although there are some negative biases in the SH. In contrast to the results for altitude at 30.5 km (right panel in Figure 4), the LP values at 15.5 km are systematically smaller than the ones from SAGE III. A discussion of this systematic difference is given in Section 4.3. A notable feature in Figure 5 is that the comparisons in the Northern Hemisphere (NH) are generally better than in Southern Hemisphere (SH). In the southern mid-latitudes (35°S-55°S), the LP retrievals show significant seasonal variations at 20.5 and 25.5 km that are not seen by SAGE. The obvious seasonal variability in the differences, with the amplitudes of winter minima was observed to be about 25% at 20.5 km, and as much as 200% at 15.5 km. The presence of erroneous seasonal variations in the OMPS/LP dataset is mostly caused by limitation of wavelength at 675 nm when observing in backscatter condition at extreme large scattering angles. The results lead us to recommend filtering LP data below 20 km with SA greater than 145°. The limitations of the LP retrievals at 675 nm will be addressed in Section 4.6."

L. 25, p.6: What do the authors mean by "limitation of 675"Âa?ËŸ

Reply: That meant limitation of the limb radiances at 675nm. We have added a new section (4.6 limitations of Wavelength) to address it.

L. 25-27, p.6: Could the authors verify their statement about the influence of Canadian PyroCb from 2D maps? Otherwise, they should qualify: "was most probably responsible (. . .)".

Reply: We have added "most probably" in the sentence as requested.

L. 27-29, p.6: The reduced data coverage in the case of SAGE III/ISS doesn't allow any relevant conclusion about a maximum limit of the disagreement between both datasets in the sense that the most prominent patterns found by LP are not observed by SAGE III/ISS. The authors should thus avoid such a quantitative, possibly misleading estimate of the agreement between both datasets.

Reply: We have removed 'up to about a factor of 4' from the sentence as requested.

L. 30, p.6-L. 2, p.7: The sentence "For SAGE, . . ., for OMPS/LP" seems to be an explanation of the statement "Under low aerosol condition". However, straylight contamination and sensitivity to small aerosol particles are two independent concepts.

Reply: We have modified the text to make this clearer: "Additionally, uncertainties in LP measurements are assumed to be 1% (Kramarova et al., 2018) and the primary source of error in LP radiance measurements is the straylight error which increases with altitude. At higher altitudes where LP radiance is small, the straylight error becomes most significant."

L. 8-9, p.7: A typical daily latitudinal coverage in a given bin (if it contains any measurement) is about 0.3◦ for SAGE III/ISS, which is very small compared to the bin latitude resolution of 5◦. The SAGE III/ISS are regularly spread over the whole longitudinal dimension, so that few points concern the region of the eruption. Did the authors check that the sampling provided by LP and mainly SAGE III/ISS, with respect to the spread of the volcanic plume, is adequate to conduct a relevant intercomparison? Otherwise, they should repeat their comparison by using a more adequate choice of bins (e.g. a more focussed region, possibly during more days). This is very important in order to distinguish the part of algorithm performance and the part of mismatch in the differences observed here. And these aspect should be at least discussed adequately.

Reply: This isn't entirely accurate. Daily SAGE latitudinal sampling widths vary by

latitude. In the tropics, the spread can be large (up to 10 degrees). The spread can be as low as less than a degree at the sampling turnover point, but the latitude of that location changes seasonally (though it's always at the higher end of the latitude range). Summarily, a 5-degree bin width may actually be narrower than the SAGE sampling in the tropics and a little wider to much wider at mid-to-high latitudes. We agree your comments about using a more adequate choice of bins (e.g. a more focussed region, possibly during more days), but we think that SAGE III has enough profiles (about 12-14 SAGE profiles in each bin) within the volcanic plume to represent it, particularly in the early period where there is not much of longitudinal spread.

L. 22-24, p.7: The authors should mention the highest altitude reached by the PyroCb, in order to provide an insight about its expected impact on the aerosol profile.

Reply: We have added text: "Within two months after injection the plume can reach up to nearly 22 km."

L. 30, p7-L.2, p.8: What is the expected impact of the wrong aerosol type (sulfate instead of carbonaceous aerosols) and possibly inadequate choice ASD in this case? Is this impact expected to be stronger in the case of LP than in the case of SAGE III/ISS? This should be discussed.

Reply: Again, SAGE doesn't need to make assumptions about the type of aerosol for the retrieval of extinction. However, LP needs to assume a refractivity index which is mainly dependent on aerosol type. We added explanatory text: "Since the LP algorithm assumes a fixed refractivity index which is dependent on aerosol type, the wrong aerosol type (sulfate instead of carbonaceous aerosols) leads to an incorrect phase function, which then produces an error in the retrieved extinction. For SAGE, this assumption is not needed, so there is less impact."

L. 17-19, p.8: I guess the 1.8 values is close to the reported averaged SAGE II values at 18 km, *in the case of reduced aerosol load*. It is anyway not representative for a high aerosol burden. This should be specified.

Reply: We thought that it was clear that 1.8 values is close to the reported averaged SAGE II values at 18 km for the same stable background aerosol loading period (2000-2005), but have added the words "in the case of reduced aerosol load" to the end of this sentence."

L. 27, p.8: "the vertical variability of stratospheric aerosol properties": the authors should be more specific.

Reply: We have changed the words to "the altitude dependency of stratospheric aerosol properties" for clarity.

L. 31, p.8: "(. . .) to see if an ASD error exists": the authors should reword this strange sentence and specify what they mean.

Reply: Thanks for pointing out this poor wording. We have corrected this wording, removing "to see if an ASD error exists," and replaced it with "to examine if the assumed ASD is reasonable".

Title of Section 4.4: The authors announce a discussion about the correlation between the extinction and the Angstrom exponent, what seems to foretell some rigourous study with appropriate calculations of the correlation between these quantities. But the reality appears to be (apparently) that (fast) conclusions are drawn from some visual examination of the similarity between two plots. The discussion falls thus short of the expectations. The methodology looks insufficient and either it should be revised, expanded or clarified, or the authors should change the title of the section. See also comment on L. 24, p.9.

Reply: The reviewer is correct that the discussion is too short to justify a separate section, so we removed the title and treating this section as the closing paragraph of Section 4.3

L. 10, p.9: What do the authors mean by "vertically smoothed SAGE III/ISS data"?

Reply: We describe our vertical smoothing procedure in Section 3.2 (Equation (2))".

We have modified the text "... derived from the vertically smoothed SAGE III/ISS data calculated using Eq.2"

L. 11, p.9: Is the SAGE extinction at 675 nm processed as previously (interpolation using 2 close spectral channels)? This should be specified.

Reply: We interpolate to get a replacement 675 nm extinction data set for SAGE III using the method described in Section 3.2. Now we are saying that we need extinction at 520 nm, which we obtain using a different interpolation method which is clearly explained in the text.

L. 14-17, p.9: The formulation is confusing. Do I understand well that the authors interpolate the aerosol extinction at 520 nm? If the interpolation makes use of a second order polynomial, why do they need 4 channels? Or are they rather fitting such polynomial? Is this approach more accurate than just deriving directly the AE using a linear fit of the aerosol extinction on a log-log scale?

Reply: We can clarify slightly by changing the beginning of the sentence to: "However, to avoid potential bias, the aerosol extinction is first interpolated to 520 nm using a second order polynomial ..." The shape appears mostly quadratic (or perhaps quartic) but it is definitely not linear across the large spectral range.

L. 17-19, p.9: This sentence is confusing and should be more accurate. The values above 2.08 found above 25 km height are not associated to the Mt. Ambae eruption, although they constitute the majority of the category of values > 2.08.

Reply: This can be further clarified to: "AE values above the baseline are possibly associated with the volcanic eruption of Mt. Ambae below 25 km (note transport from the QBO) and aerosol evaporation at higher altitudes (particularly in the tropics)."

L. 19-20, p.9: AE below the baseline (=2.08 following L. 17, p.9) are all values plotted in blue colours in Figure 11a. It is very hard to believe that all these values are easily associated to clouds. What do the authors mean? The authors should also specify

where they see the influence of PyroCb.

Reply: This can be clarified: "AE values are typically below the baseline through the mid stratosphere in non-volcanic conditions, though the smallest values in the upper troposphere are possibly associated with clouds (and a minor influence from the PyroCb in the lower stratosphere at the northern extent of the plotted data in late 2017)."

L. 21-22, p.9: I guess the authors draw this conclusion from Figure 11b. It might be useful to specify it for the sake of clarity. Furthermore, this negative bias, following Figure 11b, doesn't cover the whole stratosphere and the authors should be more specific. Finally, it seems from Figure 11b that the vertical extend of the negative bias decreases with time. Is this due to ageing of the instrument? Or do the authors have another interpretation for this general trend?

Reply: The reviewer is misinterpreting the statement. The word "bias" in this statement is not referring to differences between OMPS and SAGE aerosol extinction. This can be clarified by changing "There also appears to be an overall bias towards an AE value slightly below the baseline throughout most of the stratosphere." to "SAGE II data suggest that the overall AE throughout most of the stratosphere is slightly below the baseline value of 2.08 used by the OMPS algorithm." Further, we would not say Figure 11b suggests any kind of trend in the difference. Rather, it is more likely that the changes seen are in-line with the phase of the QBO, though perhaps that needs to be shown or at least stated that we looked into it.

L. 22-23, p.9: Where is this statement coming from? The difference between LP AE and AE derived from SAGE is not shown anywhere. Do they mean SAGE III/ISS AE illustrated in Figure 11a? Otherwise, they need to show appropriate results to support their statement.

Reply: This is part of the foundation of this entire Section and thus illustrates where much of their confusion is coming from. OMPS/LP uses a flat value of 2.08, which is stated clearly in the text. Figure 11a uses this value as the zero on the color scale and

so it shows the difference between AE from SAGE and AE used for OMPS by virtue of the color rather than by value. We added text: "Red colors in Figure 11a thus represent SAGE III/ISS AE values larger than the OMPS/LP AE value, while blue colors represent AE values smaller than the OMPS/LP value"

L. 24, p.9: There is of course a strong correlation between AE and ASD, but these quantities are still different. Furthermore, visually, the altitude range takes roughly blue colours in similar regions of both plots in Figure 11 (say, the "lower stratosphere"), but the details of the time evolution of this altitude range is different in Figs. (a) and (b) and the correlation between the negative values (the various blue tones) cannot be assessed without an adequate calculation. Therefore, saying that there is "a clear correlation" is absolutely premature (this apparent correlation might be purely fortuitous), and should be removed or clearly qualified.

Reply: We revised the text as "The similar structure between AE variations shown in Figure 11a and the extinction differences shown in Figure 11b suggest that the use of an altitude-dependent ASD in the LP retrieval is a significant component of the observed extinction differences, although other factors may also contribute."

L. 25-27:, p.9: Where is this statement coming from? The only results shown about southern mid-latitudes are the ones in Figure 5, and I don't see how they could lead to these results. The authors should bring the necessary developments and/or explanations to support their conclusion, or remove this sentence.

Reply: The reviewer is correct that Figure 11 only shows the tropics. We removed the sentence and replace it with the following: "Section 4.5 gives further discussion of how the assumed LP phase function and LP measurement geometry can result in latitude-dependent variations in extinction differences with SAGE III."

L. 10, p.10: The extinction units are missing.

Reply: We have added missing "m-1".

L. 11, p.10: What do the authors mean by "wavelength limitations"?

Reply: This is a very good point. The reviewer also mentioned this point several times through the review (see Comment on L. 25, p.6; L. 24, p.6; L. 12-15, p.10; L. 11, p.10 and Figure 13). As indicated in the manuscript, the OMPS algorithm uses a single wavelength at 675 nm which results uncertainties in the southern mid-latitudes at lower altitude. The uncertainties may be reduced by doing LP aerosol retrieval at longer wavelengths.

To address this question, we added a new section (new Section 4.5) with a new figure (Figure 14) as follows: "4.5 Wavelength Impact on OMPS/LP Aerosol Sensitivity In Figures 5 and 12, the comparison shows asymmetry between the hemispheres below 20.5 km, with much better agreement in the NH than in the SH, and OMPS/LP extinction values are significantly biased at southern mid-latitudes below 20 km due to erroneous seasonal variations in the OMPS/LP dataset. That suggests the LP measurements are more sensitive to aerosols in the NH and less sensitive to those in the SH, especially at lower altitudes. Here we shall examine the sensitivity of the LP radiances to aerosol. As mentioned before, the LP V1.5 algorithm uses OMPS/LP radiances at a single wavelength at 675 nm to retrieve extinction profile. This wavelength was selected primarily to minimize aerosol-related errors in the ozone retrieval and to reduce stray light contamination (Loughman et al., 2018). However, as indicated in Sections 4.1 and 4.4, it is difficult to retrieve reliable aerosol extinction below 20 km in the SH due to lack sensitivity to aerosol. Figure 13 shows an example of aerosol weighting functions at 675 nm for three latitudes. The aerosol weighting function, which determines how the calculated radiance (I) at a given wavelength changes with a change in aerosol extinction (k), is denoted by: $(\partial \ln(I))/(\partial \ln(k))$ (3) The derivatives are calculated for all altitudes for a change at each tangent height, and each curve in the figure shows the sensitivity of the radiance at a given tangent height to extinction perturbations of a 1 km layer at a range of altitudes. It can be seen that the sensitivity to aerosol varies with latitude and altitude. The LP radiance at 675 nm is most sensitive to

the aerosol extinction over the 20-30 km altitude range in the tropics and the northern mid-latitudes, but less sensitive to aerosol in the southern mid-latitudes. This behavior is consistent with the results shown in Figures 5 and 12 that the LP retrievals are in better agreement with the SAGE data in the NH than those in the SH. In the limb scattering technique, on the other hand, the aerosol signal in the limb radiance at a given tangent height is roughly proportional to the product of the aerosol extinction in that layer and the PF at the tangent point. OMPS/LP is installed in a fixed orientation relative to the S-NPP spacecraft. Therefore, southern mid-latitudes are observed at backscattering geometries whereas NH observations are carried under forward scattering conditions (see Loughman et al., 2018, Figure 2). Aerosol scattering phase function is at least fifty times smaller for OMPS limb viewing in SH than in NH. Due to the variation of the PF with latitude and season, the LP observations are most sensitive to aerosols in the NH winter and least sensitive to those in the SH. LP scattering angles typically vary between 15 and 165°. For the selected wavelength of 675 nm and the assumed ASD, the PF has much smaller values at larger scattering angles (see Chen et al., 2018, blue line in Figure 2), corresponding to the larger solar azimuth angles at southern latitudes. This leads to a smaller relative contribution of aerosol scattering with respect to Rayleigh scattering. At extreme large scattering angle (>145°) where Rayleigh scattering is high and the value of the aerosol phase function is close to its minimum, the LP radiances at 675 nm are nearly insensitive to aerosol at lower altitudes. Therefore, the SA upper limit (= 145°) should be used to filter the data. Outside of this SA range, LP extinction values will tend to bias the retrieved result, as revealed in Figure 5. This cloud explain the presence of erroneous seasonal variations in the OMPS/LP dataset. Since the sensitivity to aerosol increases at longer wavelengths (Taha et al., 2011), the uncertainties in the southern mid-latitudes at lower altitude may be reduced by using longer wavelengths. Figure 14 illustrates the sensitivity of the limb radiances to the aerosol extinction at three wavelengths for latitude at 60 °S (solar zenith angle SZA = 70°). The sensitivity to aerosol is seen to increase with increasing wavelength. The increased sensitivity of the limb radiances to aerosol at longer wavelengths is partly due

to the fact that the Mie scattering from aerosol particles does not decrease as rapidly with wavelength as the Rayleigh scattering from air molecules (Bourassa et al., 2007)."

L. 11, p.10 and Figure 13: The concept of "aerosol weighting function" is never defined in the text and should be appropriately explained. Reply: We added an explanatory paragraph in the new section (Section 4.6). Please see Reply to Comment on L. 11, p.10.

L. 12-15, p.10: Could the authors shortly explain the reason for the different sensitivity to aerosol in the northern mid-latitudes and tropical latitudes, and the southern midlatitudes?

Reply: We add text: "OMPS LP is installed in a fixed orientation relative to the S-NPP spacecraft. Therefore, southern mid-latitudes are observed at backscattering geometries whereas NH observations are carried under forward scattering conditions. Aerosol scattering phase function is at least fifty times smaller for OMPS limb viewing in SH than in NH." Also please see Reply to Comment on L. 11, p.10.

L. 4-5, p.11: See comments on L. 24, p.6 and L.24, p.9. I don't agree about the conclusion on the robustness of the measured extinction variability.

Reply: The "robust" isn't used in a rigorous manner. We have removed this strong statement.

L. 8-10, p.11: Measurement uncertainties are never discussed nor quantitatively mentioned before.

Reply: This is discussed now by adding a new sentence in the end of Section 4.3 (see Reply to Comment on L. 30, p.6-L. 2, p.7).

L. 19, p.11: "differing" is not an adequate term. What is a "differing good agreement"?

Reply: We changed the text to "differing good agreement and much larger differences above 30 km and below 19 km".

L. 4-7, p.12: These aspects are discussed in a paper by Malinina et al. (2019), amt, 3485-3502. It seems appropriate to cite this paper.

Reply: The work of Malinina et al. (2019) has been mentioned and referenced in the manuscript.

L. 7-9, p.12: "Lower sensitivity" of what? Which kind of uncertainties are the authors talking about? To which specific (and different) concepts do the authors refer by "lower sensitivity" and "reduced retrieval accuracy"? The increased extinction uncertainty above 28 km and below 19 km mentioned in L. 5, p.9 seems to refer to retrieval uncertainty, while it is not clear if the uncertainty cited in L. 15 and 18, p.10 and linked to "noise amplification" refers to instrumental noise or retrieval noise. The discussion on uncertainty assessment is clearly insufficient.

Reply: It meant lower sensitivity of LP radiances at 675 nm to aerosols. The extinction uncertainty above 28 km and below 19 km mentioned in L. 5, p.9 is to refer to retrieval uncertainty. Here we are talking about the remaining discrepancies in the comparison are possibly attributed to the inherent uncertainties associated with different measurement techniques which affect the instrument sensitivity to aerosols. The "lower sensitivity of LP radiances at 675 nm to aerosols" is now discussed in Section 4.5.

We have also rewritten the paragraph about uncertainty in the Summary and Conclusions: "Our sensitivity analysis shows that the ASD error propagates into extinction uncertainty as much as 13%. This suggests that a dynamic model for ASD is needed to accurately retrieve aerosol extinction profiles. There are three other possible causes for the discrepancies. First, most of the remaining discrepancies in the comparison are possibly attributed to the inherent uncertainties associated with the measurement techniques. The differences in the measurement techniques affect the instrument sensitivity to aerosols (Malinina et al., 2019). While SAGE III/ISS uses the solar occultation measuring technique which is self-calibrating and derives extinction directly, OMPS/LP

employs the limb scatter technique where retrieved extinction depends on instrument calibration and tangent height registration as well as the inversion algorithm and its several assumptions including the assumed aerosol microphysics. Uncertainties in these assumptions can also affect the LP extinction product..."

L. 10, p.12: What do the authors mean by "straylight contamination is more obvious"?

Reply: We modified the sentence as "The limb radiance is also more susceptible to additive straylight contamination at high altitudes, where absolute radiance values are smaller."

L. 11, p12: The concept of "discrepancy" looks strange.

Reply: "a larger random discrepancy" is replaced by "large difference".

L. 11-12, p.12: This sentence should be qualified. It is not clear whether the authors consider the association between the large discrepancy between LP and SAGE, and the presence of clouds as obvious (in my opinion, it is not), or as a working hypothesis.
Reply: We qualified the discrepancy as "as much as 60%"

Abstract and summary/conclusions: It seems appropriate and important to explicitly mention the presence of erroneous "seasonal variations" in the OMPS-LP dataset in the abstract and in the summary and conclusions, since this behaviour risks to induce misinterpretations in future works.

Reply: Thank you, this is an important point. We add a paragraph to the Abstract that reads "On the other hand, we find erroneous seasonal variations in the OMPS/LP dataset, which usually exists below 20km in Southern Hemisphere due to the lack of sensitivity to particles when scattering angle is greater than 145°."

We also add a paragraph to the end of the Summary and Conclusion that reads "Another significant finding in this study is that the comparison shows asymmetry between the hemispheres below 20.5 km, with better agreement in the NH than in the SH, and erroneous seasonal variations at southern in the OMPS/LP dataset with an estimated

uncertainty of a factor up to two. The reason for this is identified as that the LP radiances at 675 nm are nearly insensitive to aerosol at extremely large scattering angles. This problem can be solved by using longer wavelengths to retrieve aerosol extinction. In the meantime, we recommend filtering LP data below 20 km with scattering angle greater than 145°. When scattering angle exceeds this limit the LP algorithm starts to give obviously wrong results."

Technical comments General remark: The authors make use of both SAGE II and SAGE III/ISS datasets. Sometimes, they refer to "SAGE". They should be more specific.

Reply: Fixed.

General remark: I guess the correct spelling is "mid-latitude" instead of "midlatitude".

Reply: Fixed throughout the paper.

L. 22, p.3: The units should be corrected (microns instead of m).

Reply: The word document that we see is $\mu$m, not m. In the pdf version, however, it was changed. Now we reconvert word to pdf and see the same.

L. 22, p.6, L23, p.8, Caption Figure 11, l.11, p.9, etc.: "extinction" is a physical parameter and should be singular. If the authors want to use a plural form, they should use "extinction values". Please check the whole document.

Reply: Agreed. We changed "extinctions" to "extinction values" throughout the paper.

L. 30-31, p.6: The expression "The AE is quite scattered" seems improper to me. AE is a physical property.

Reply: Agreed. We changed "AE" to "AE value".

L. 30, p.6: I don't think that the formulation "smaller aerosols" is appropriate. Aerosol is a substance in suspension in the air. The authors should use the term "particle" that

can be associated to the concept of size.

Reply: "aerosols" has been changed to "particles".

L. 9, p.9: missing punctuation. Caption Figure 11: It is confusing that the labels (a) and (b) are after the corresponding part of the caption. The authors should put the labels first.

Reply: Fixed.

L. 29, p.10: The correct expression for "Chappius" is "Chappuis bands". The authors should use the correct one.

Reply: Corrected.

L. 4, p.11: Which time series? Please be specific.

Reply: We have added text "during the period of June 2017∼ May 2019".

L. 6, p.11: I guess there is only one retrieval? And "Impact" seems more appropriate than "impacts".

Reply: Fixed.

L. 9, p.12: duplicated word.

Reply: Done.

---

## Author Comment (AC2) · 30 Jan 2020

Reply to Reviewer 2 Z. Chen et al. zhong.chen@ssaihq.com

We thank Reviewer #2 for useful comments. Below we answer the reviewer's concerns and make the necessary corrections to the paper and supplement.

The Stratospheric Aerosol Layer is an important component of the earth atmosphere through its impacts on climate and stratospheric ozone especially after large volcanic eruptions. Chen et al. (2019) evaluate the OMPS/LP stratospheric aerosol product using the SAGE III/ISS observations. They found a good agreement (+/-25%) between OMPS/LP and a modified version of the SAGE III V5.1 data between 20 and 28 km. OMPS/LP and SAGE III/ISS data are analyzed after a moderate volcanic eruption and

extreme fire reaching the stratosphere to highlight the contribution of those events on the stratospheric aerosol extinction. Finally, the sensivity of the aerosol retrieval to the assumed size distribution is also investigated at the end of the study. I have a number of major concerns about this paper before it can be published in AMT.

1- Introduction. The introduction does a very poor job in explaining why this work is important and why the stratospheric aerosol layer should be studied. I suggest the authors to do a literature overview of this topic to explain.

Reply: We add a paragraph to the beginning of the Introduction that reads "The Stratospheric stratospheric aerosol layer is an important component of the Eearth's atmosphere through its impacts on climate and stratospheric ozone physico-chemistry (Vernier et al., 2011; Ridley et al., 2014; Bingen et al., 2017). The stratospheric aerosol layer was first observed by Junge in 1960 (Junge et al., 1961). Stratospheric aerosols that mainly originate from volcanic sources are from sulfur dioxide and carbonyl sulfide, which are both oxidized to sulfuric acid (Kremser et al., 2016). Recent measurements show that the background stratospheric aerosol layer is variable rather than constant, and the that changes in the stratospheric aerosol layer have caused recent warming rates (Solomon et al., 2011), indicating that it is important to monitor the stratospheric aerosol layer over the long term. The importance of possible changes in the background stratospheric aerosol layer lets to the analysis of each volcanically quiescent period (Deshler et al., 2006). For the considered period from January 1979 through the end of 2004, the variability of stratospheric aerosol layer is explored with measurements from space‐based instruments such as SAGE II (Thomason et al., 2008), CALIPSO (Winker et al., 2010), GOMOS/ENVISAT (Vanhellemont et al., 2010), SCIAMACHY (von Savigny et al., 2015), OSIRIS/Odin (Bourassa et al., 2007), SAGE III/ISS (Chu et al., 1998) and OMPS/LP (Loughman et al., 2018)."

2- Novelty of this study? As mentioned by the authors, a precedent paper was published last year to evaluate the new OMPS/LP aerosol product (V1.5) with the SAGE III/ISS data. This study aims to extend this analysis with one more year observations

but do not further explain the scientific justifications for providing this update. What is so different between this paper and Chen et al. (2018) ? This is not justified with the publication of a new algorithm or new version of the OMPS dataset so why is it important to publish this?

Reply: The present study is a more comprehensive evaluation of the OMPS/LP aerosol product than previous one (Chen et al., 2018). In previous study, we were unable to evaluate the magnitude of the aerosol variability at the temporal scale at which this comparison is conducted. While the previous comparison used 7 months of SAGE data (V5.0), this comparison uses the latest version (V5.1) of SAGE III/ISS observations for the period from 2017 to 2019. In this study, as noted in the manuscript, the agreement and discrepancy between OMPS/LP extinction profiles and SAGE III/ISS observations are quantitatively assessed, some problems of aerosol retrievals from LP are identified and explained. Furthermore, a smoothed+interpolated method for SAGE profile is employed, which allows us minimize the uncertainty due to vertical resolution issue and to avoid ozone contamination. This evaluation will play an important role in future improvements of the LP aerosol dataset. We add the following text to the end of Introduction: "Our analysis of SAGE III/ISS data specifically addresses possible biases with OMPS/LP results arising from differences in vertical resolution and possible ozone contamination."

3- Justification for using CARMA in April 2012. An important part of the retrieval is the assumption of size parameters into the radiative transfer model to infer the aerosol extinction at 675 nm from OMPS/LP. The description of the algorithm (section2) provides the basis to understand how the extinction is inferred. A gamma size distribution is used to fit size distribution from the CARMA aerosol module running with GEOS model in April 2012. I have several questions associated with this approach: - Why do you use a Gama function to fit the model data? Bi-Lognormal distributions have been commonly used to fit stratospheric aerosol data such as those observed by the University of Wyoming for more than 30 years (Deshler et al., 2003) - It's rather strange to

use one month of model data as an input to constrain a retrieval algorithm. Moreover, the caveat is that the satellite output data will not able to be used by modelers using CARMA-GEOS since they are not independent to each other.

Reply: We realized that "using CARMA in April 2012" here is miscommunicated. It should be "the LP started observations from April 2012". We have corrected this wording by removing "and observations in April 2012" in the abstract. The reason why we use a Gama function to fit the CARMA model data has been intensively explained by Chen et al., (2018). While bimodal lognormal distributions is commonly used for the in situ instruments (Deshler et al., 2003; Deshler, 2008), most satellite instruments, such as OSIRIS, SCIAMACHY and OMPS/LP v1.0, use a unimodal lognormal distribution (Rault, et al., 2013; Damadeo et al., 2013; Rieger et al., 2014; von Savigny et al., 2015; Malinina et al., 2018). Chen et al. (2018) fitted four bi-modal lognormal distributions to the same underlying OPC measurements measured by the University of Wyoming. The resulting size distributions gave very similar values of Angstrom exponent, but significantly different phase functions due to the lack of information in the OPC data gap region (between $0.01\mu$m - $0.1\mu$m). The fitting results indicated that the abundance of smaller particles can significantly affect the phase function, and therefore affect our LP retrievals.

In the Introduction, we add a paragraph capturing these concerns. "We use the CARMA data to take advantage of CARMA's large range of particle size information, and a gamma size distribution represents a significantly better fit to the CARMA data than a lognormal distribution. In a recent study, Nyaku et al. (2019) confirms that the CARMA model agrees with the Wyoming OPC measurements." We hope this provides a justification for the use of the gamma function.

4- Modification/improvement of the SAGE III/ISS official product. In order to correct an apparent issue with the 675 nm extinction coefficient from SAGE III/ISS due to interference with ozone, the authors developed a new algorithm to interpolate the 675 nm channel data using 449 and 756 nm. Without providing further validation of this

technique, the authors acknowledge that the new retrieved 675 nm coefficient from SAGE III is used for comparison and to some extent validate OMPS-LP. I think the approach is questionable here: Without further validation of the new retrieved 675nm extinction coefficient from SAGE III/ISS, you assume that it will be your new reference to compare and validate OMPS-LP. I think OMPS-LP should first be validated/compared with the official product from SAGE III/ISS at 675 nm before transforming the SAGE III data.

Reply: We did compare LP data with the official product from SAGE III/ISS at 675 nm in the beginning of this work and shared our initial results with SAGE science team. As the SAGE science team pointed out, bias in ozone result in a bias in the official aerosol product from SAGE III/ISS at 675 nm. To avoid ozone contamination, we developed a new algorithm to interpolate the 675 nm channel data using 449 and 756 nm per their suggestion. Our results show that the differences between the original and interpolated profiles below 27 km appear to be minimal. The SAGE team started an ozone validation paper (currently under review in JGR:A) where they brought this up in the context of how apparent biases in the aerosol spectrum likely originate from how ozone is retrieved and the impact on the ozone product (though the cause is ozone and the effect is aerosol). This paper does discuss the retrieval algorithm a little more. We now provide this reference: Wang, H.J., R. Damadeo, D. Flittner, N. Kramarova, G. Taha, S. Davis, A.M. Thompson, S. Strahan, Y. Wang, L. Froidevaux, D. Degenstein, A. Bourassa, W. Steinbrecht, K. Walker, R. Querel, T. Leblanc, S. Godin-Beekmann, D. Hurst, and E.J. Hall, E.: Validation of SAGE III/ISS Solar Ozone Data with Correlative Satellite and Ground Based Measurements, JGR Atmos., 2020, in review.

Comments: 1) P1-l16: "has been flying". I believe that this expression could be improved in a scientific publication

Reply: We have modified the sentence as " ... has been taking limb-scattered measurements from April, 2012–present."

2) P1-l29: "high degree of correlation". Quantify here.

Reply: We quantified this by changing this sentence to the following: "and the differences between the two instruments vary from 0% to $\pm$25% depending on altitude, latitude and time."

3) P1-L32: "systematically lower: : :" You are not measuring the same air masses so the different between the two instruments are expected to be high at the tropopause or below.

Reply: Agreed. The tropopause is a very dynamical and complicated area. We have added text in the end of Section 3.2: "Furthermore, the large different between the two instruments are expected at lower altitudes near and below the tropopause because of larger variability in the transport of air masses." and Section 4.3: "Near or below the tropopause, big disagreements between the two instruments can also be expected due to the variability of the transport of air masses."

4) P1-L34: "altitude dependent..." Not only altitude but also latitude.

Reply: Here we emphasized the fact the actual aerosol size distribution is altitude dependent. We prefer to keep unchanged.

5) P2-L3: "cloud contamination": That is of the main issue, which is poorly discussed in this paper.

Reply: In Section 3.2, we have added the following text: "Cirrus cloud contamination could be another issue in the lower stratosphere below about 19 km. Clouds appear as discontinuities of the limb radiance vertical profiles, which will tend to bias the retrieved result. Although most clouds are detected and filtered from the LP retrievals, it is not always possible to completely eliminate cloud contamination."

6) P2-L8: " (Ridley et al., 2014) Â′ n. There is a very poor review of the available literature on this topic. This should be improved.

Reply: We have added more references. Please see Reply to Comment 1.

7) P2-26: "has become operational..." What does it mean here?

Reply: The sentence has been modified to be "The version 1.5 OMPS/LP aerosol products are now being processed routinely"

8) P2-L27: "..A more comprehensive..": Does it justify another publication on OMPS-LP?

Reply: We added "more" here.

9) P4-L10: "The SAGE III/ISS developed..": Something is missing between SAGE III/ISS and the verb. It does not read well.

Reply: We add missing "which".

10) P5-L10: "Cloud height rejected: : :": How is cloud top height inferred from OMPS-LP? \

Reply: This has been made clearer with: "The current cloud detection algorithm (Chen et al., 2016) detects cloud top height from the OMPS/LP measurements using the spectral dependence of the vertical gradient of radiances at 675 and 868 nm. Cloud top height is identified when the gradient difference increases above 0.15. All LP data below the cloud top height were rejected because extinction changes abruptly at cloud top."

11) P6-L19: "Figure 11..": You should remove a reference to a figure that you do not explain at this stage of the paper.

Reply: We have removed "(see Figure 11)".

12) P7/Figure7. Figure 7 does not really highlight nicely the emergence of new strato-spheric aerosol layers before and after the Ambae eruption and the fire in Canada. I would rather suggest producing an anomaly plot before and after each event.

Reply: This is a nice suggestion. We replot data in Figure 7 by adding anomaly results. The paragraph in the end of Section 4.2 has been rewritten to be "Figure 7 highlights the emergence of new stratospheric aerosol layers before and after the fire event in Canada in late 2017. Both instruments clearly captured the stratospheric aerosol per-turbation triggered by the reported PyroCb. We chose July 6, 2017 as the "before" case (Figure 7a), based on the results shown in Figure 5, and then calculated the difference for LP and SAGE separately using November 9, 2017 as the "after" case (Figure 7b) to show the anomaly results. As expected, Figure 7c shows large positive extinction anomalies of up to 140 % below 22 km. The comparison also shows good agreement between LP retrieved extinction profiles and observations from SAGE III within $\pm$20 % at 18-28 km."

13) P7-10. How sure are you that the corresponding increase in extinction is associated with this eruption? Provide reference or further analysis to make your point.

Reply: We provide here three references: "Vernier et al., 2011; Kremser et al., 2016; Bingen et al., 2017."

14) P7-L23. "were produced.." use a better verb than "produce" here.

Reply: This sentence has been rewritten as the following: "The occurrence of PyroCbs triggered by intense wildfires in British Columbia, Canada were reported on August 12, 2017".

15) P8-L8. "..for the main aerosol layer.." What do you mean by "main layer", the Junge layer? The stratospheric aerosol reservoir in the tropics? Be more accurate so that the reader can understand what you're talking about.

Reply: We dropped "for the main aerosol layer" as it not accurate.

16) P8-L27-28: "The results: : :" I do not understand this sentence. Please rephrase and improve.

Reply: The sentence as rewritten as: "that when taking into account the altitude dependency of stratospheric aerosol properties, the LP algorithm performance improves at most levels."

17) P9-L18: "are easily associated.." It does not read well in English. Please improve.

Reply: We replaced "easily" by "possibly"

18) P9-L19: You need to include references here.

Reply: Reply: We added five references: Schuster et al., 2006; Kremser et al., 2016; Rieger et al., 2018; Malinina et al., 2019.

19) P12-L12: "..broken clouds.." What do you mean by broken clouds, cirrus clouds?

Reply: " broken clouds" is replaced by "thin cirrus clouds".

---

## Referee Report (RR1)

**Evaluation of OMPS/LP Stratospheric Aerosol Extinction Product Using SAGE III/ISS Observations, by Zhong Cheng et al.: Review of the revised version**

**General Comments**

The authors improved significantly the quality of the paper by adding uncertainties, by being more accurate and specific in their argumentation, and by removing some uninformative statements. The new sub-section 4.5 "Wavelength impact on OMPS/LP Aerosol Sensitivity" brings an interesting insight, especially for readers less familiar with the field of limb scattering and with the OMPS instrument in particular.

Overall, I believe they address successfully the comments expressed by the reviewers. I formulate some additional minor issues that should be addressed before publication.

**Specific comments**

L. 5, p.10: It is not clear to me what the relationship is between Figure 5 and the selection of the "before" and "after" cases.

Title new sub-section 4.5: I am wondering if the choice of the word "impact" is optimal, since "impact" has a connotation of "result of an action". The use of "dependence" or "influence" might be more appropriate.

L. 7, p.15: "both" instead of "the both" ?

L. 22, p.15: As already mentioned in the previous referee report, the expression "differing good agreement" sounds strange to me and might need revision.

P. 1-16: As already mentioned in the previous referee report, in many places, the authors use the word "SAGE" and should be more specific (i.e. use "SAGE III/ISS") to avoid any confusion.

L. 16, p.16: The quantity "60%" is mentioned for the first time here, in the conclusions. The authors should either mention from where they inferred this estimate (maybe Figure 3?) or better, should provide this estimate also in the text at the place it comes out from the discussion.

Figure 13: The mention of the latitude for the right panel is not consistent in the picture (62°N) and in the caption (59°N).

Reply to comment on Figure 3 and L. 1, p6: In Bingen et al., 2004, the median radius found in the lower stratosphere is larger than at higher altitudes (see Figure 2). Concerning the case of other aerosol types, a suitable reference is Brühl et al., Atmos. Chem. Phys., 18, 12845–12857, https://doi.org/10.5194/acp-18-12845-2018, where the authors investigate the radiative forcing of different kinds of aerosols including dusts and organics, and mention the importance of sea salt increasing the particle size in the UTLS due to water uptake.

Reply to comment on L. 8-9, p.7: I am not sure that the authors interpret rightly my comment. The point is that a volcanic plume probably only covers a very limited area of the corresponding bin of 5° latitude x 360° longitude. Therefore, the measurement locations found in the bin for SAGE III/ISS and OMPS-LP may either be situated inside, or outside the area covered by the plume. If, as an example,

the events corresponding to one instrument fall by chance inside the plume and the ones corresponding to the other instrument fall by chance outside the plume, no relevant information will be available to assess the agreement between both sensor measurement datasets. It is this adequation of the coverage that has to be checked before comparing the data from SAGE III/ISS and OMPS-LP.

---

## Author Response (AR2)

We thank the editor and reviewers for their comments. We have responded to each comment and made appropriate changes to the manuscript. Authors' responses to the specific comments are below in black.

**Editor comments:**

I have now received two reviews of the revised version of your manuscript and both reviewers found that the paper has improved significantly. I agree with the reviewers that a minor revision would further improve the paper and ask you to consider and include the specific reviewer comments.

**Thank you for your decision. We have produced a minor revision of the paper that includes the specific reviewer comments.**

Anonymous Referee #1:

General Comments

The authors improved significantly the quality of the paper by adding uncertainties, by being more accurate and specific in their argumentation, and by removing some uninformative statements. The new sub-section 4.5 "Wavelength impact on OMPS/LP Aerosol Sensitivity" brings an interesting insight, especially for readers less familiar with the field of limb scattering and with the OMPS instrument in particular. Overall, I believe they address successfully the comments expressed by the reviewers. I formulate some additional minor issues that should be addressed before publication.

Specific comments

L. 5, p.10: It is not clear to me what the relationship is between Figure 5 and the selection of the "before" and "after" cases.

The fire plume was encountered in August, 2017. We chose July 7, 2017 as the "before" case and November 10, 2017 as the "after" case.

We have removed 'based on the results shown in Figure 5' from the sentence because there is no relationship between Figure 5 and the selection.

Title new sub-section 4.5: I am wondering if the choice of the word "impact" is optimal, since "impact" has a connotation of "result of an action". The use of "dependence" or "influence" might be more appropriate.

We have replaced "impact' with 'dependence'.

L. 7, p.15: "both" instead of "the both"? We have replaced "the both' with 'both'.

L. 22, p.15: As already mentioned in the previous referee report, the expression "differing good agreement" sounds strange to me and might need revision.

**This was a typo. We have removed "differing good agreement" from the sentence.**

P. 1-16: As already mentioned in the previous referee report, in many places, the authors use the word "SAGE" and should be more specific (i.e. use "SAGE III/ISS") to avoid any confusion. We agree. The word 'SAGE' has been replaced everywhere by 'SAGE III/ISS' or 'SAGE II'.

L. 16, p.16: The quantity "60%" is mentioned for the first time here, in the conclusions. The authors should either mention from where they inferred this estimate (maybe Figure 3?) or better, should provide this estimate also in the text at the place it comes out from the discussion.

**The reviewer is correct that the quantity "60%" is estimated from Figure 3. We have added this estimate to the last paragraph of section 3.2.**

Figure 13: The mention of the latitude for the right panel is not consistent in the picture ( $62^{\circ}N$ ) and in the caption ( $59^{\circ}N$ ).

**Typo fixed.**

Reply to comment on Figure 3 and L. 1, p6: In Bingen et al., 2004, the median radius found in the lower stratosphere is larger than at higher altitudes (see Figure 2). Concerning the case of other aerosol types, a suitable reference is Brühl et al., Atmos. Chem. Phys., 18, 12845–12857, https://doi.org/10.5194/acp-18-12845-2018, where the authors investigate the radiative forcing of different kinds of aerosols including dusts and organics, and mention the importance of sea salt increasing the particle size in the UTLS due to water uptake.

**We agree your comments about aerosols with larger particle size in the UTLS. We have added the following text:**

"In addition, sea salt can also increase the particle size in the UTLS due to water uptake (Brühl et al., 2018)."

Reply to comment on L. 8-9, p.7: I am not sure that the authors interpret rightly my comment. The point is that a volcanic plume probably only covers a very limited area of the corresponding bin of 5° latitude x 360° longitude. Therefore, the measurement locations found in the bin for SAGE III/ISS and OMPS-LP may either be situated inside, or outside the area covered by the plume. If, as an example, the events corresponding to one instrument fall by chance inside the plume and the ones corresponding to the other instrument fall by chance outside the plume, no relevant information will be available to assess the agreement between both sensor measurement datasets. It is this adequation of the coverage that has to be checked before comparing the data from SAGE III/ISS and OMPS-LP.

The reviewer is correct regarding the evaluation of such plumes shortly after the initial injection. In this study, we are not attempting to make comparisons based on the early observations of a plume event. The SAGE III/ISS sampling of any specific latitude has frequent time gaps ranging from a few weeks to almost two months, as shown in Figure 2. The initial smoke plume from the Canadian pyroCb event was observed at 65°-70°N, beyond the viewing coverage of SAGE III/ISS, and then was rapidly transported to lower latitudes and eastward. By the time that SAGE III/ISS observations became available at 50°N in September 2017, the plume had encircled the Northern Hemisphere. In this situation, a zonal mean analysis should be a reasonable characterization of the plume contribution to the overall atmosphere. Similarly, the first SAGE III/ISS measurements at 0°-5°N following the Ambae eruption did not occur until 17 August 2018, approximately 3 weeks later. OMPS LP data show enhanced extinction values at 15-20 km at all longitudes by this time, indicating significant zonal transport since the initial eruption. We have therefore used a zonal mean analysis for this comparison as well. The text in Section 4.2 has been revised to clarify our rationale for this approach.

**Anonymous Referee #2:**

Most of the points raised by the referees have been addressed. However, I think the main point of referee #1 was that the zonal comparisons performed in the paper were insufficient to resolve fresh volcanic plumes. The authors have relaxed the conclusions to specify the analysis is applicable to background conditions, but I think this is insufficient:

1. several smaller eruptions and two large wildfires are present during the OMPS-LP period making these an important component of the record.

2. Coincident comparisons are a fairly standard approach that could be readily applied to the volcanic and wildfire enhanced regions in the overlap period. If this is to be an extension of the 2018 paper, then I think this analysis is warranted. At least it should be discussed why this is not appropriate, or not needed.

**Please see our response to Reviewer #1 regarding our approach to these points.**

References have been improved, would also suggest changing:

"...stratosphere (UTLS), particularly in the tropics, the presence of thicker aerosol layers may have an impact on the retrieved extinction levels (Kremser et al. 2016)." Kremser is a poor reference for retrieval specifics, and primary literature should be cited.

We have included two more references: "Bourassa et al., 2012; Fromm et al., 2014l;".

**Evaluation of OMPS/LP Stratospheric Aerosol Extinction Product Using SAGE III/ISS Observations**

Zhong Chen1,2, Pawan K. Bhartia2, Omar Torres2, Glen Jaross2, Robert Loughman3, Matthew DeLand1, Peter Colarco2, Robert Damadeo4 and Ghassan Taha5

1Science Systems and Applications, Inc., Lanham, MA, USA
 2NASA Goddard Space Flight Center, Greenbelt, MA, USA

[revised manuscript text omitted]

  S. Strahan, Y. Wang, L. Froidevaux, D. Degenstein, A. Bourassa, W. Steinbrecht, K. Walker, R. Querel, T. Leblanc, S. Godin-Beekmann, D. Hurst, and E.J. Hall, E.: Validation of SAGE III/ISS Solar Ozone Data with Correlative Satellite and Ground Based Measurements, JGR Atmos., 2020, in review.

30

10

35

Figure 1. Time series of SAGE II Angstrom Exponent (AE) derived from the aerosol extinction coefficients measured at 525 nm and 1020 nm for altitudes of 30 km (blue), 25 km (green) and 20 km (red). This figure shows SAGE II version 7.0 data for the 0–10° N latitude band during the period 1986 - 2005. While the Pinatubo eruption in 1991 produced a significant decrease in AE, the smaller volcanic eruptions such as Ruang/Reventador in 2002 and Manam in 2005, visible in the extinction time series (not shown), did not appreciably affect AE values. The AE values appear to stabilize after 2000, suggesting that a background state exists. The AE is quite scattered at 30km compared to at lower altitudes, which is related to reduced quality of the aerosol retrieval for low aerosol loading.

---

## Author Response (AR3)

Dear Associate Editor,

We thank you for your useful comments and for efforts you have been spending for editing our manuscript. We have produced a minor revision of the paper that includes your comments. Below we answer your concerns and make the necessary corrections to the paper.

*Page 2, line 12: "Stratospheric aerosols that mainly originate from volcanic sources are from sulfur dioxide and carbonyl sulfide" I think something is missing here?*
**Reply:** We rewritten the sentence as
 "Stratospheric aerosols that mainly originate from volcanic sources are described as liquid droplets composed of a mixture of the sulfuric acid ($H_2SO_4$) and water ($H_2O$)"

*Page 3, line 9: "it's" -> "its"*
**Reply:** Fixed.

*Page 4, line 23: "Other time periods known to have low aerosol loading (e.g. 1989- 1990) show lower values of AE in the SAGE II dataset." It's not entirely clear what this statement refers to. Looking at Fig. 1 and 30 km, the AE in 1989/1990 shows a maximum.*
**Reply:** We have added the words "between 20 and 25 km" to the sentence for clarity.

*Page 5, line 12: "The SAGE III/ISS which developed by"*
*"which was developed" or just "developed" ?*
**Reply:** Fixed.

*Page 7, line 10: "SAGE III/SS"*
**Reply:** Fixed.

*Page 8, line 18: "While the SAGE III/ISS algorithm does not make any assumptions"*
*This is certainly correct, but perhaps it would better to write "does not require any assumptions" ?*
**Reply:** We have replaced "make" by "require" as suggested.

*Page 8, line 21: "uncertainties in LP measurements" -> "uncertainties in LP radiance measurements"?*
**Reply:** We have added "radiance' after 'LP'.

*Page 15, line 12: "profilers" -> "profiles"*
**Reply:** Fixed.

*Page15, line 20: "The results shown that the LP retrievals"*
**Reply:** Fixed.

*List of references: There are many inconsistencies in the list of references (spaces between initials etc.), please check the AMT style guide and adjust the references accordingly.*
**Reply:** Done.

*Caption Fig. 5: "angels" -> "angles"*
**Reply:** Fixed.

*Fig. 7: I think something is wrong here. Looking at panels a) and b), the differences below 19 km or so are larger for LP than for SAGE, right? But in panel c) the anomaly is larger for SAGE. This appears inconsistent. Please correct me, if I'm wrong.*
*Please also mention how the extinction anomaly is defined exactly. What is the reference profile?*
**Reply:** You are correct. We have fixed this issue by changing the color of the two lines in panel c). We also updated the caption of Fig. 7c to clarify our definition:
 "c) extinction anomaly (defined as deviation from the before case)"

*Fig. 8: Is it possible, that the two lines are mixed up? The behavior of the lines is inconsistent with intuition and also with Fig. 9.*
*Larger particles - associated with lower AE - will have a more pronounced forward scattering peak, i.e. PF(AE=1.8) - PF(AE2.08) should be positive for a scattering angle of zero. However, this not the case in Figure 8. Similarly, PF(AE=2.3) – PF(AE2.08) should be negative for a scattering angle of zero.*
*In addition, the signs of the curves in Fig. 8 appear to be inconsistent with the differences between the curves in Fig. 9. In Figure 8 PF(AE=2.3) is smaller than PF(AE=2.08) for scattering angles between 60 and 90 deg, which would imply larger aerosol extinction for the AE=2.3 case (to produce the same limb radiance). However, this is not the case in Fig. 9.*
*These issues would disappear, if the two lines in Fig. 8 are mixed up or if the differences are actually determined the other way around (i.e., B – A instead if A – B). Perhaps I'm missing something?*
*Please also specify, what value is used as a reference value to determine the relative differences in Figure 8.*
**Reply:** Thank you for catching this. This was due to a label typo in Fig. 8. We have updated the labels in Fig. 8 – now these issues disappear. As indicated in the figure caption and labels, the reference value, i.e. the baseline value, is 2.08.

*Caption Fig. 14, line 1: "675 nm (left)"*
*The panel says "508 nm" Same caption, line 5: "508 nm and 675 nm" ?  Which wavelength is shown?*
**Reply:** We updated the caption Fig. 14 to indicate that 508 nm, 745 nm and 868nm are shown. Thank you.